

# Aerosol emissions estimation with POLDER

Athanasios Tsikerdekis[1,2,3], Otto P. Hasekamp[1], Nick A. J. Schutgens[2], Qirui Zhong[2]

[1]SRON Netherlands Institute for Space Research, Leiden, the Netherlands
[2]Department of Earth Science, Vrije Universiteit Amsterdam, 1081 HV Amsterdam, the Netherlands
[3]now at: Royal Netherlands Meteorological Institute (KNMI), De Bilt, the Netherlands

*Correspondence to*: Otto Hasekamp (O.P.Hasekamp@sron.nl), Athanasios Tsikerdekis (thanos.tsikerdekis@knmi.nl)

**Abstract.** We apply a Local Ensemble Transform Kalman Smoother (LETKS) in combination with the global aerosol climate model ECHAM-HAM to estimate aerosol emissions from POLDER-3/PARASOL observations for the year 2006.
We assimilate Aerosol Optical Depth at 550mnm ($AOD_{550}$), Ångström Exponent for 550nm and 865nm ($AE_{550-865}$) and Single Scattering Albedo at 550nm ($SSA_{550}$) in order to improve modeled aerosol mass, size and absorption simultaneously. The new global aerosol emissions increase to 1419 $Tg \cdot yr^{-1}$ (+28%) for dust, 1850 $Tg \cdot yr^{-1}$ (+75%) for sea salt, 215 $Tg \cdot yr^{-1}$ (+143%) for organic aerosol and 13.3 $Tg \cdot yr^{-1}$ (+75%) for black carbon, while the sulfur dioxide emissions increase to 198 $Tg \cdot yr^{-1}$ (+42%) and total deposition of sulfates to 293 $Tg \cdot yr^{-1}$ (+39%). Organic and black carbon emissions are much higher
than their prior values from bottom up inventories with a stronger increase in biomass burning sources (+193% and +90%) than in anthropogenic sources (115% and 70%). The evaluation of the experiments with POLDER (assimilated) and AERONET as well as MODIS Dark Target (independent) observations shows a clear improvement compared to the ECHAM-HAM control run. Specifically based on AERONET the global mean error of $AOD_{550}$ improves from -0.094 to -0.006 while $AAOD_{550}$ improves from -0.009 to -0.004 after the assimilation. A smaller improvement is observed also in
$AE_{550-865}$ mean absolute error (from 0.428 to 0.393), with a considerably higher improvement over isolated island sites over the ocean. The new dust emissions are closer to the ensemble median of AEROCOM I, AEROCOM III and CMIP5 as well as some of the previous assimilation studies. The new sea salt emissions get closer to the reported emissions from previous studies. Indications of a missing fraction of coarse dust and sea salt particles are discussed. The biomass burning changes (based on POLDER) can be used as alternative biomass burning scaling factors for the GFAS inventory distinctively
estimated for organic carbon (2.93) and black carbon (1.90), instead of the recommended scaling of 3.4 (Kaiser et al. 2012). The estimated emissions are highly sensitive to the relative humidity due to aerosol water uptake, especially in the case of the sulfates. We found that ECHAM-HAM, like most of the GCMs that participated in AEROCOM and CMIP6, overestimated the relative humidity compared to ERA-5 and as a result the water uptake by aerosols, assuming the kappa values are not underestimated. If we use the ERA-5 relative humidity, sulfate emissions must be further increased, as
modeled sulfate AOD is lowered. Specifically, over East Asia, the lower AOD can be attributed to the underestimated precipitation and the lack of simulated nitrates in the model.



# 1 Introduction

A prominent uncertain component in aerosol modeling are the aerosol emissions. The uncertainty of aerosol emissions
enhances the unpredictability in the simulated aerosol concentration and optical properties (Textor et al., 2007) as well as
aerosol radiative effect and forcing (Myhre et al., 2013; Yoshioka et al., 2019). A bottom-up estimate of anthropogenic
aerosol emissions is usually coming from the integration of known sources of information across different economic sectors,
such as power, industry, transport and residential (Zhang et al., 2009). These bottom-up techniques are very useful since they
provide a first-guess estimate of aerosol emissions, but emission differences in source attribution (power, industry,
residential) may lead to very different simulated aerosol concentrations (Saikawa et al., 2017).

Natural aerosol emissions like dust and sea salt are estimated in aerosol models through different schemes by using wind
speed as well as land or ocean characteristics (Grythe et al., 2014; Long et al., 2011; Tegen et al., 2002). A large fraction of
the natural emissions diversity can be attributed to differences in the modeling approaches. Emission schemes can differ in
the parameterization of source strength as a function of wind (Grythe et al., 2014; Textor et al., 2007), the simulated wind
themselves (Textor et al., 2007), the simulated size spectrum of the emitted particles (Kok et al., 2021; Textor et al., 2006),
the simulated size grouping in each model (e.g. modes or bins) (Gliß et al., 2021), the implementation of spatial filters where
dust emission sources can dynamically change based on vegetation (Wu et al., 2020). Also note that large differences in the
simulated natural emissions can emerge by simply using a different horizontal resolution in the same model (Guelle et al.,
2001; Laurent et al., 2008). In addition, note that the physically relevant scale (about 1m to several km) where dust emissions
can vary is not captured by the current horizontal resolution of global climate models (Kok et al., 2021).

Emissions from biomass burning emissions are based on satellite measurements that are related to burned area and use
emission factors to convert the burned dry matter into emissions of aerosol and gas species (GFED4; (Van Der Werf et al.,
2017), active fire count (FINN1.5; (Wiedinmyer et al., 2011) or fire radiative power (QFED2.4; (Darmenov & da Silva,
2015), FEER1.0; (Ichoku & Ellison, 2014) and GFAS; (Kaiser et al., 2012). It has been shown that different emission factors
may contribute to the diversity between these emission inventories, but differences in the dry matter have also been reported
for North America fires (Carter et al., 2020), which is one of the main reasons that the fire detection and/or fire burden area
inventories do not align with fire radiative power inventories (Van Der Werf et al., 2017). In addition, strong inter-annual
differences as well as regional diversity are observed between the datasets, with a fairly good agreement over the Amazon
and a quite high disagreement over Africa and boreal North America (Carter et al., 2020).

The global dust emissions relative diversity (usually quantified as the ratio of the standard deviation to the mean (Schutgens
et al., 2020) for the multi-model ensemble of CMIP5 is 87% (Wu et al., 2020), for AEROCOM I is 73% (Huneeus et al.,
2011), while for several simulations from a single model with diverse emission scheme settings is 61% (Miller et al., 2006).



The sea salt emission relative diversity is 97% based on several different sea salt emission functions (Grythe et al., 2014), for global estimates within the range of 1200 to 20000 Tg·yr$^{-1}$ as proposed by (Lewis & Schwartz, 2004). The emission relative diversity from biomass burning based on six emission datasets is 76% for organic carbon and 82% for black carbon (Pan et al., 2020). Consequently, the global emissions of aerosol from natural sources, such as desert (dust), oceans (sea salt) and

non-anthropogenic biomass burning (organic and black carbon) is at best higher than 60%, hence there is a lot of room for improvement.

The anthropogenic emissions differences between inventories for aerosol or aerosol precursor is considerably lower than the one of natural emissions. In Lee et al. (2013) lower OC and BC uncertainty was used for fossil fuel compared to biomass

burning emissions as well as lower $SO_2$ uncertainty for fossil fuel compared to volcanic emissions. The emission diversity estimated by multiple anthropogenic emission inventories, as the ratio of highest to lowest anthropogenic global emissions, showed that it is lower than 20% for BC and $NO_x$ and lower than 42% for $SO_2$ after the year 2000 (Granier et al., 2011). The anthropogenic aerosol and aerosol precursor gas emissions relative diversity over large areas is significantly lower, but note that these different emissions inventories are constructed using very similar information and methods and are not

independent from each other (Granier et al., 2011). Based on four emission inventories over eastern China for 2006, the emissions relative diversity (using the mean in the denominator) of $SO_2$, ammonia ($NH_3$), OC and BC is 5%, 18%, 12% and 16% respectively (Chang et al., 2015). Note that this diversity is based on yearly means, hence the day to day variability and relative diversity among these emission inventories can be higher. Further, the sector attribution of emissions can be quite different in each dataset, which can affect the uncertainty of emissions on the regional level (Saikawa et al., 2017).


These high emissions differences for modeled fluxes of dust and sea salt as well as differences in fluxes in emission inventories for the other aerosol species led to the popularization of top-down method that combine simulated aerosol information from a model and retrieved aerosol information from satellites (Chen et al., 2018, 2019; Dubovik et al., 2008; Escribano et al., 2017; Huneeus et al., 2012; Jin et al., 2019; Pope et al., 2016; Schutgens et al., 2012; Sekiyama et al., 2010;

X. Xu et al., 2013). The simulated aerosol state in the model is produced using background emissions which are either prescribed from emission inventories (anthropogenic aerosols and biomass burning respectively) or interactively calculated through emission schemes (dust and sea salt aerosols). In addition, the uncertainty of the assimilated observations and the uncertainty in the background emissions need to be specified.

Note that most of these studies estimate new emissions based on the assimilation of Aerosol Optical Depth (AOD), some may include also Ångström Exponent (AE), while very few assimilate absorption observations, like Absorption Aerosol Optical Depth (AAOD) or Single Scattering Albedo (SSA) (Chen et al., 2018, 2019). By not including observations of measurements related to size and absorption, the estimated emission may be misrepresented as it has been shown for the estimated aerosol mixing ratio in (Tsikerdekis et al., 2021), where several data assimilation experiments were conducted



with different combinations of observations from the POLDER instrument. The multi-wavelength and multi-viewing-angle photopolarimetric measurements of POLDER contains more information about the scattered solar radiation compared to single-viewing measurements (Hasekamp & Landgraf, 2007; Mishchenko & Travis, 1997), hence POLDER is an ideal tool for obtaining accurate aerosol microphysical and optical properties, which potentially can provide a more accurate estimation of emissions, as suggested in Schutgens et al. (2021).


Although aerosol emissions are critically uncertain, other factors can affect the uncertainty in modeled aerosol concentration and optical properties. One of these factors is the aerosol water uptake in models that can considerably increase the simulated AOD diversity (Gliß et al., 2021). The misrepresentation of water uptake can have a huge impact, since the condensed water over dry aerosol particles may contribute up to 70% of the total AOD globally (K. Zhang et al., 2012). During the

AEROCOM I phase substantial diversity among the model was attributed to differences in the modeled water uptake (Kinne et al., 2006). A recent study evaluated the scattering enhancement factor of 10 Earth system models based on 22 ground based in situ measurements (Burgos et al., 2020). The scattering enhancement factor for a certain wavelength ($\lambda$) is the ratio of light scattering coefficient under wet (RH=85%) to dry (RH=40%) conditions, which describes the increase of aerosol scattering due to the wet growth of particles under different RH conditions. The results showed that the models tend to

overestimate scattering enhancement factor as an ensemble mean by 15%, though the differences from model to model were quite substantial. The inter-model differences were attributed to different assumptions in kappa and contrasting growth for low RH (RH<40%) conditions between the models. Further it was suggested that lower kappa values should be used in the models for organics and sea salt and considerable differences were found between the models for light scattering enhancement factor under relatively dry conditions (RH<40%). Although this study was very insightful, the scattering

enhancement factor analyses cases with the same RH conditions but potentially a very diverse aerosol load, since the low and high RH conditions may have occurred in different times and dates for each model and for the observations. In our study we assume that kappa is correct for our experiments and investigate how a biased RH may influence aerosol water growth, their optical properties and aerosol estimated emissions by the data assimilation system.

The effect of a biased RH, which can dramatically affect the simulated aerosol optical properties, received little attention. The current horizontal resolution of Global Climate Model (GCMs), which for the majority of AEROCOM III and CMIP6 models is between 1° and 2° (Gliß et al., 2021; Z. Xu et al., 2021), cannot resolve humidity's small scale processes, thus they are parameterized through cloud schemes (Lin, 2014). Because of this, biases in the simulated humidity can accumulate in GCMs. The specific humidity of the CMIP5 ensemble is overestimated over mid-latitudes throughout the troposphere when

compared to Atmospheric Infrared Sounder (AIRS) (Tian et al., 2013). Further, the majority of the CMIP6 model (12 out of 18), overestimate relative humidity at 850hPa in all seasons compared to ERA5 (Z. Xu et al., 2021).





In the present study we estimate the aerosol emissions of dust (DU), sea salt (SS), organic carbon (OC), black carbon (BC), sulfates (SO4) and precursor gasses emissions for sulfates like sulfur dioxide (SO2) and dimethyl sulfide (DMS) for the year 2006. Our method implements a Local Ensemble Transform Kalman Smoother (LETKS) which was introduced in our preceding work (Tsikerdekis et al., 2022). It combines POLDER observations, that were retrieved by the algorithm developed at the Netherlands Institute for Space Research (SRON), with the aerosol information simulated by ECHAM-HAM. We assimilate AOD550, AE550-865 and SSA550 in order to simultaneously account for the correction of aerosol mass, size and absorption (Tsikerdekis et al., 2021). In addition, we conduct sensitivity and data assimilation experiments using the relative humidity of ERA5 (instead of ECHAM-HAM) for the water uptake process, to quantify the effect it has on aerosol optical properties and the estimated emissions. Section 2, presents the retrieved observations from POLDER and the model ECHAM-HAM. The observations and emissions uncertainties are discussed. Section 3 briefly describes the LETKS and provides an overview of our experiments. Section 4 includes the evaluation results of our experiments against POLDER and independent (AERONET and MODIS) observations as well as the new estimated emissions along with the reported emissions from previous studies.  In addition, we quantify the effect of a biased high RH on aerosol optical properties and emissions.

## 2 Data

### 2.1 Aerosol Observations (POLDER)

POLDER-3 is an instrument that can measure light intensity and polarization properties for up to 16 viewing angles and multiple wavelengths (0.44 to 1.02μm). In addition, the multi-angle multi-wavelength photopolarimetric measurements have the ability to differentiate scattering of cloud droplets from aerosol particles, thus the exclusion of cloud contaminated pixels is possible (Stap et al., 2015). The instrument was part of the Polarization and Anisotropy of Reflectances for Atmospheric Sciences coupled with Observations from a Lidar (PARASOL) micro-satellite, which was active during 2004 to 2013.

The POLDER observations that were used in this study were retrieved by an algorithm developed at SRON - Netherlands Institute for Space Research, which fits a radiative transfer model (Hasekamp & Landgraf, 2005; Schepers et al., 2014) to the multiangle photopolarimetric measurements of POLDER to derive aerosol optical properties corresponding to a bi-modal aerosol size distribution. We use the global bimodal product, which is the only product available globally, but note that a regional 10 mode achieved higher accuracy for AOD and similar performance for SSA when compared to AERONET for retrievals over land (Fu & Hasekamp, 2018). The retrieved properties for a fine and a coarse particle mode are the effective radius, the effective variance, the column number concentration as well as the real and imaginary part of the refractive index for each mode (Hasekamp et al., 2011, 2019; Lacagnina et al., 2015; L. Wu et al., 2015). Using the abovementioned aerosol parameters, for the two modes, Aerosol Optical Depth (AOD), Angstrom Exponent (AE), Absorption optical Depth (AAOD) and Single Scattering Albedo (SSA) can be calculated. The aerosol optical properties of POLDER retrievals demonstrate



good agreement with either ground based (AERONET) or satellite (Ozone Monitoring Instrument; OMI) retrievals for the
year 2006 (Hasekamp et al., 2011; Lacagnina et al., 2015, 2017; Stap et al., 2015).

In the present study aggregated (1° × 1°) POLDER data are used in the assimilation for the year 2006. POLDER uncertainty
for each assimilated observable was estimated for several POLDER AOD550 bins based on an AERONET evaluation and is
presented on Appendix A. Note that POLDER AE550-865 over Sahara is biased high based on AERONET, thus these
observations were not assimilated (see Appendix A). A more detailed description of the use of POLDER data in our
assimilation system can be found in (Tsikerdekis et al., 2021) and details on the SRON POLDER retrieval algorithm can be
found in (Fu et al., 2020; Fu & Hasekamp, 2018).

**2.2 Aerosol Model (ECHAM6-HAM2)**

The aerosol climate model ECHAM6-HAM2 (mentioned as ECHAM-HAM onward) is used to simulate the meteorological
and aerosol state of the atmosphere. The model consists of two parts, the general circulation model ECHAM6, developed at
the Max Planck Institute for Meteorology (MPI-M) in Hamburg, Germany (Stevens et al., 2013), and the second version of
the Hamburg Aerosol Model (HAM2) (Stier et al., 2005; Tegen et al., 2019; K. Zhang et al., 2012). Aerosols are simulated
in seven unimodal lognormal particle size distributions (modes), four of them are the hydrophilic Nucleation, Aitken,
Accumulation and Coarse while three of them are the hydrophobic Aitken, Accumulation and Coarse. Each mode may
contain one or more (internally mixed) aerosol species, namely dust (DU), sea salt (SS), organic carbon (OC), black carbon
(BC) and sulfates (SO4) (Vignati et al., 2004). Currently the model does not simulate aerosol nitrates. The cloud and aerosol
optical properties are computed using Mie Theory and derived from lookup tables (Tegen et al., 2019) using the prognostic
concentrations of aerosol tracers (Schultz et al., 2018).

All aerosol species are emitted, transported, deposited and take part in aerosol-radiation interactions (scattering and
absorption) as well as aerosol microphysical processes (e.g. nucleation, coagulation, aerosol water uptake and cloud
activation) (Schutgens & Stier, 2014; K. Zhang et al., 2012). The natural aerosol types (DU, SS) are introduced to the
atmosphere by utilizing the simulated information of wind and certain surface and ocean characteristics. Other aerosol
species (OC, BC) or aerosol precursor gasses (SO2, DMS) that are emitted from both natural (biomass burning or biogenic
emissions) and anthropogenic sources (e.g. industry and transport) use predefined emission inventories (K. Zhang et al.,
2012). Specifically, anthropogenic emissions are derived from 14 different sectors. Each sector may include one or more
aerosol types or aerosol precursors (Schultz et al., 2018; Tegen et al., 2019). A more detailed description of the model can be
found in our preceding works (Tsikerdekis et al., 2021).

Aerosol water uptake is the process of condensing water vapor on the surface of aerosol particles. This process affects
aerosol's size, deposition, atmospheric lifetime and optical properties. Thus, it is crucial to simulate it accurately in aerosol





models. In ECHAM-HAM water uptake is simulated by a semi-empirical water uptake scheme (O'Donnell et al., 2011) that approximates the enhancement of particle size (growth factor; gf) based on (Petters & Kreidenweis, 2007). Based on this

scheme the growth of aerosol particles depends on the relative humidity (RH), the dry particle radius (Dp), the kappa parameter (κ) which is distinctive for each aerosol species and determines its hygroscopicity as well as the Kelvin term (A) that is a temperature dependent constant (O'Donnell et al., 2011). In order to enhance computational efficiency this equation is solved offline and organized in look up tables where the aerosol growth factor can be determined for specific RH, Dp, k and A conditions in each grid cell of the model. Kappa expresses the volume of water that is associated with a unit volume of

dry particles (Petters & Kreidenweis, 2013) and the higher it gets the more soluble the aerosol species is. In ECHAM-HAM the kappa is fixed for each species, specifically the kappa for SS, SO4, and OC is equal to 1.00, 0.60 and 0.06 respectively. DU and BC  are considered insoluble (kappa=0). The most decisive parameter of the above, that influences the growth factor of soluble particles (high κ) the most, is RH. Hence, in this study we conduct experiments where RH from ERA5 is explicitly used for the water uptake of aerosols in ECHAM-HAM to quantify its effect on the simulated aerosol optical properties.

Further, this option is adopted in a data assimilation experiment to quantify the effect of RH on aerosol emission estimation.

## 3 Methods

### 3.1 The Local Ensemble Transform Kalman Smoother

The Local Ensemble Transform Kalman Smoother (LETKS) is used to estimate aerosol emission fluxes. This method was previously used by Schutgens et al. (2012) for aerosols emission estimation and earlier for CO2 emission estimation

(Bruhwiler et al., 2005; Peters et al., 2005; and Feng et al., 2009). A detailed description of LETKS can be found in Tsikerdekis et al. (2022) where the method and the code was tested for aerosol emission estimation using SPEXone synthetic measurements in Observing System Simulation Experiments (OSSEs). Here the main components of the method are discussed.

The system estimates perturbation to the background emissions and assumes that these perturbations remain constant over 2 days. The emission perturbations are estimated using assimilation cycles, where each cycle consists of a background and an analysis step. The background step produces the required background information based on a 8-day ($\Delta T_b$) forward simulation of ECHAM-HAM driven by a priori ("background") emissions. The analysis step assimilates all the available POLDER observations within the last 2 days ($\Delta T_s$) of the forward simulation and estimates the "analysis" emissions for the

last 6 days  ($\Delta T_a = \Delta T_b - \Delta T_s$) of the forward simulation.

At the end of each assimilation cycle the estimated analysis emissions of the previous cycle serve as background emissions for the next cycle, time is shifted forward equal to $\Delta T_s$ days and the background and analysis steps are repeated. Note that with this setup several assimilation cycles overlap in time, thus the estimated emissions (estimated in batches of 2 days) are





affected by observations of the current and subsequent days. Specifically, the emissions of a day may be affected by observations of the same day and of the five subsequent days. This iterative design ensures that observations close to the sources along with observations away from the sources (e.g. an aerosol plume created by particles emitted several days earlier), will be both used to correct the emissions.

The assumed background emissions are uncertain. The uncertainty of the emissions is represented with an ensemble of 32 simulations where emissions are perturbed. The perturbation is conducted by multiplying the emissions with spatially correlated perturbations (see subsection 3.2 on Tsikerdekis et al., 2021). The spatial correlation length scale of the perturbations is approximately 25° omnidirectionally, except for DU perturbations over Sahara where the spatial correlation length is zero (perturbations from grid to grid are uncorrelated). The zero spatial correlation length for DU over Sahara was
chosen after conducting several data assimilation experiments with different correlation length values and evaluating them in terms of AOD$_{550}$ (not shown). Each perturbation set is uniquely generated for every perturbed parameter and ensemble member. In each grid cell, the mean of the background distribution of emission scaling factor for the first cycle is equal to 1, while for all subsequent cycles is set equal to the analysis distribution mean of the previous cycle (see prior correction subsection in Tsikerdekis et al. 2022). In each grid cell, the standard deviation of the background distribution, which
represents the uncertainty of the emissions, is distinct for each perturbed parameter and is further discussed in Appendix B.

New emission estimates are obtained by estimating scaling factors based on the assimilated observations by solving the Kalman filter equations:

$$x_a = x_b + P_a \cdot H^T \cdot R^{-1} \cdot (y - H \cdot x_b) \,, \tag{1}$$

$$P_a = (I + P_b \cdot H^T \cdot R^{-1} \cdot H)^{-1} \cdot P_b \,, \tag{2}$$

where $\mathbf{x_b}$ is the background state vector and includes emission perturbations for each species (DU, SS, OC, BC and SO4). Different perturbations are used for each optically relevant mode (Aitken, Accumulation, Coarse) and biomass burning (BB) or fossil fuel (FF) contributions. Specifically, the emissions that are distinctively perturbed and estimated (11 in total) by the assimilation system are shown in Table 1. The perturbation of sulfate precursor gasses (SO2 and DMS) used the same
perturbations as SO4. $\mathbf{x_a}$ is the analysis state vector, containing the retrieved emission scaling factors based on the assimilated observations ($\mathbf{y}$). $\mathbf{P_b}$ and $\mathbf{P_a}$ are the covariance matrices corresponding to the background and analysis state vector, respectively. The observational uncertainties are represented by the error covariance matrix $\mathbf{R}$. We assume R to be diagonal (i.e. correlations between observational errors are assumed to be zero always). The observation operator $\mathbf{H}$, translates the emission perturbations (x) to the simulated observations (H·x) and it is entirely handled by the model
(emission, transport, deposition, aerosol processes and optical properties code). T stands for the matrix transpose operator.



## 3.2 The Local Ensemble Transform Kalman Smoother

All experiments are conducted using the model ECHAM-HAM for the year 2006. The experiments use 31 vertical sigma-hybrid levels from the surface up to 10hPa (Troposphere only simulations), a T63 horizontal resolution of 1.875° x 1.875° and are nudged to ERA5 surface pressure as well as to vorticity, divergence, and temperature for all vertical levels.


$CTL_{ECHAM}$ is an ECHAM-HAM run without data assimilation and with default settings, while $DAS_{ECHAM}$ is the data assimilation experiment where the emissions are optimized based on measurements by POLDER. In addition, we conducted an experiment with an identical setup to $CTL_{ECHAM}$, but with lower horizontal resolution (T31; 3.75° x 3.75°).

$CTL_{ERA5}$ quantifies the effect of the underestimated relative humidity in ECHAM compared to ERA5 on aerosol optical properties. $CTL_{ERA5}$ uses the relative humidity of ERA5 for aerosol water uptake. Note that this modification affects only the simulated aerosol optical properties in ECHAM-HAM, while the simulated water cycle (precipitation and evaporation) of the model remains unaltered. $DAS_{ERA5}$ quantifies the effect of overestimated relative humidity profile to the aerosol emission estimation.

**4 Results**

## 4.1 The Local Ensemble Transform Kalman Smoother

All experiments were evaluated against the assimilated observations (POLDER) and independent observations (AERONET and MODIS). In both cases there is a significant improvement in all the aerosol optical properties in the $DAS_{ECHAM}$ experiment, except $AE_{550-865}$ over some land areas where the error increases, and can possibly be attributed to the relatively
high observational uncertainty for $AE_{550-865}$ (FigureA 1).

In Figure 1 the experiments $CTL_{ECHAM}$ and $DAS_{ECHAM}$ are compared to the assimilated POLDER observations for the year 2006. $CTL_{ECHAM}$ exhibits a strong underestimation in AOD over the biomass burning regions over the Tropics (Amazon, Central Africa and Indonesia) and Siberia that are dominated by organic and black carbon aerosols, as well as over arid
environments dominated by dust (Sahara, Middle East and Taklamakan/Gobi deserts). $AOD_{550}$ is overestimated over south-eastern China, where aerosol load is very high (POLDER $AOD_{550}$ is higher than 0.6) and composed mostly of sulfates, as well as over open water bodies, where aerosol load is low and dominated by sea salt. The $CTL_{ECHAM}$ $AOD_{550}$ per species is depicted in FigureS 1. The assimilation of POLDER observation ($DAS_{ECHAM}$) reduces the $AOD_{550}$ global Mean Error (ME) from -0.08 to -0.03 and the Mean Absolute Error (MAE) from 0.10 to 0.06, which shows that ECHAM-HAM can better
match the observations with adjusted emissions. Note that local improvement of $AOD_{550}$ for certain regions is even greater.





The AE$_{550-865}$ is a good proxy for aerosol size. High and low values of AE$_{550-865}$ relate to an aerosol load with more fine and more coarse particles, respectively. POLDER AE$_{550-865}$ is high over biomass burning and highly polluted regions, dominated mainly by OC, BC and SO$_4$, while is low over the ocean and deserts where the aerosol load is primarily composed of DU and SS (Figure 1d). In CTL$_{ECHAM}$ AE$_{550-865}$ is underestimated over the Sahara and middle East and eastern China while
overestimated over the ocean, Siberia and American continent. The estimated emissions by DAS$_{ECHAM}$ improve the AE$_{550-865}$ difference over the ocean and there is a significant improvement over China. The remaining high differences of AE$_{550-865}$ over land can be attributed to the high uncertainty of POLDER AE$_{550-865}$ over land. In FigureS 2 the yearly mean uncertainty of POLDER is depicted along the MAE of the 3-hourly differences between the Experiments – POLDER. The remaining MAE of the 3-hourly differences in DAS$_{ECHAM}$ (FigureS 2c) are on the same level as POLDER uncertainty (FigureS 2a),
which means that POLDER AE$_{550-865}$ over land are too uncertain to further adjust emissions. Further, sensitivity studies show that even when the biomass burning emitted particles size is altered aggressively in ECHAM-HAM, AE is not affected much (Zhong et al., 2022), which indicates that the emission changes may be less sensitive to the assimilation of AE$_{550-865}$ compared to AOD$_{550}$. The global MAE for AE$_{550-865}$ is reduced from 0.34 in CTL$_{ECHAM}$ to 0.27 in DAS$_{ECHAM}$.

The AAOD$_{550}$ highly correlates with BC aerosol load, which is the species that contributes up to 80% of the total absorption globally, followed by DU (16%) and OC (4%) (FigureS 3). POLDER AAOD$_{550}$ peaks over tropical Africa and Sahel, where large biomass burning fires are active during the fire (dry) season. Fairly high values of absorption are also observed over the Amazon basin for the same reason. Further, high AAOD$_{550}$ values are depicted over the northern and western coastline of Australia. Medium values of AAOD$_{550}$ are visible over eastern United states, Europe and eastern China, that are related to
anthropogenic emissions. POLDER depicts high AAOD$_{550}$ values also over high altitude regions (Schutgens et al., 2021), like the Rocky Mountains, the Andes, the Himalaya, Zagros mountain range in Iran, Hijaz mountain range in Saudi Arabia as well as the highlands in Ethiopia. Over these high elevation areas there are hardly any BC or DU sources, thus these values might be a product of retrieval errors related to surface elevation.

The AAOD$_{550}$ in CTL$_{ECHAM}$ is mostly underestimated globally. A pronounced underestimation is evident over Tropical Africa, which relates to the low BC emissions of the emission inventory GFAS (v1.0) we use. Typically, the biomass burning emissions of GFAS for black and organic carbon are multiplied with a scaling factor of 3.4 to obtain a similar AOD observed by MODIS (Kaiser et al., 2012; Veira et al., 2015). Here this scaling factor is not applied in order to let our data assimilation system estimate new scaling factors based on POLDER observations, distinctively for OC and BC emissions.
DAS$_{ECHAM}$ has considerably smaller differences from POLDER globally and especially over the Tropics. The global MAE for SSA is reduced from 0.0106 in CTL$_{ECHAM}$ to 0.0077 in DAS$_{ECHAM}$.

The experiments are also evaluated with independent observations that are not assimilated. The scatterplots in Figure 2 depict the evaluation of POLDER as well as the POLDER collocated CTL$_{ECHAM}$ and DAS$_{ECHAM}$ against AERONET. All



AERONET sites were collocated with the closest grid cell in one 1 x 1 resolution on a 3-hourly basis. Cases where multiple stations belonged on the same grid cell and had observations at the same time, were averaged. A similar analysis for non-collocated to POLDER evaluation with AERONET for $CTL_{ECHAM}$ and $DAS_{ECHAM}$ is provided in FigureS 4 for all AERONET stations as well as in Figure 3 and Figure 4 for selected AERONET stations representative for SS and BC, respectively.


The ME and MAE improves in $DAS_{ECHAM}$ experiments compared to $CTL_{ECHAM}$ for all variables, except the $AE_{550-865}$ ME. The satellite $AE_{550-865}$ is overestimated compared to AERONET by 0.096, which partially can contribute to the increase of the $AE_{550-865}$ ME in $DAS_{ECHAM}$. Further, the unchanged high $AE_{550-865}$ in the model is observed over land (Figure 2f) where the observational uncertainty of POLDER $AE_{550-865}$ is high (greater than 0.45) for most $AOD_{550}$ bins (FigureA 1).


The uncertainty of POLDER observations is based on an evaluation with AERONET (see Appendix A). POLDER $AE_{550-865}$ errors spread against AERONET (Figure 2d) are similar to the $CTL_{ECHAM}$ errors spread against AERONET (Figure 2e). Notably the POLDER $AAOD_{550}$ errors spread against AERONET (Figure 2g) is even greater than the $CTL_{ECHAM}$ errors spread against AERONET (Figure 2h). Despite this, there is a small improvement in MAE for both observables and a clear

improvement on AAOD550 bias where the ME goes from -0.009 in $CTL_{ECHAM}$ to -0.004 in $DAS_{ECHAM}$.

The improvement of $AE_{550-865}$ and $AAOD_{550}$ compared to AERONET after data assimilation is much more clear if we focus on AERONET stations in regions where the difference between $CTL_{ECHAM}$ and $DAS_{ECHAM}$ is large. This is mostly in regions with strongly modified SS and BC emissions, respectively. To investigate this improvement, an evaluation for selected

stations is depicted in Figure 3 and Figure 4. In Figure 3 four stations that are located in isolated islands over the ocean were selected in order to capture the changes of $AE_{550-865}$ due to the adjusted SS emissions. In all cases the $CTL_{ECHAM}$ overestimates $AE_{550-865}$. After the adjusted emissions the $AE_{550-865}$ is improved with a reduction in ME of about 0.1 or higher (except Midway Island). In Figure 4 four regions with biomass burning and anthropogenic BC emissions were selected to study the changes of $AAOD_{550}$. In all cases the underestimation of $AAOD_{550}$ in $CTL_{ECHAM}$ improves after the adjusted

emissions, especially in the sites over the biomass burning regions (Sahel stations and Mongu station), but also in regions with anthropogenic sources of BC (Europe and India). Similar improvement is observed for $SSA_{550}$ over the same regions (FigureS 5)

From previous work (Tsikerdekis et al. 2021) we know that assimilating $AOD_{550}$ along with $AE_{550-865}$ and $SSA_{550}$ results in a

considerable $AOD_{550}$ improvement with a small improvement on size and absorption. Assimilating only $AOD_{550}$ results in a considerable $AOD_{550}$ improvement, small improvement in aerosol size while having a very negative effect on the aerosol absorption. Our findings here confirm the importance of assimilating information on size and absorption in addition to AOD.


It is important to note that future polarimeter instruments such as SPEXone and 3MI are expected to yield better retrievals (Hasekamp, Fu et al., 2019) and hence the potential to estimate aerosol emissions better (Tsikerdekis et al. 2022).


In addition, we evaluate the effect of the assimilation against MODIS Collection 5 Dark Target (Sayer et al., 2014) at 1° x 1° resolution. Specifically, we use a specialized version of MODIS designed for assimilation, which was corrected based on four years of AERONET observations (Hyer et al., 2011; Shi et al., 2011; J. Zhang & Reid, 2006). Figure 5 depicts the MODIS $AOD_{550}$ for the year 2006 along with the differences of $CTL_{ECHAM}$ and $DAS_{ECHAM}$ from MODIS. Before the

assimilation the model biases against MODIS follow a similar pattern to the biases observed against POLDER, with an underestimation of $AOD_{550}$ over land (notably over biomass burning regions and Sahel) and an overestimation over ocean. After the assimilation the negative bias over land is corrected, but the overestimation over ocean remains. The ME and MAE improve from -0.032 and 0.061 in the $CTL_{ECHAM}$ to 0.015 and 0.050 in the $DAS_{ECHAM}$ experiment. Further, we conduct a similar analysis to Figure 2 but with the MODIS data. The scatter plots in Figure 6 depict the collocated points between

MODIS and AERONET for 2006, which are more than five times greater in number compared to POLDER. Similarly, before the assimilation a negative bias is observed which is corrected after the assimilation with a reduction of the spread of the errors as well. Specifically, the ME is reduced from -0.063 to 0.009 and the MAE from 0.132 to 0.118.

### 4.2 Aerosol emission estimation from POLDER

The yearly emissions for several aerosol species are shown in Figure 7. Dust and sea salt particles in the coarse mode

dominate the total mass of aerosols, followed by sulfates and sea salt in the accumulation mode and organic carbon emissions. Note that sulfate total deposition is used as a proxy for sulfate formation in the atmosphere. SO2 emissions are primarily concentrated over the Northern hemisphere, mainly over North America, Europe, India and Southeast Asia. Black carbon total mass is very low globally (although very important for aerosol absorption, see FigureS 3d) and concentrated over biomass burning regions and densely populated areas where high anthropogenic emissions occur.


The relative changes of yearly aerosol emissions because of the assimilated POLDER observations are depicted in Figure 8. Grid cells with emissions lower than the global median value in each species are masked out (grey), to focus on areas where aerosol emissions are not too low. Overall, emissions increase for all species (except sea salt accumulation mode), which coincides with the large underestimation of both $AOD_{550}$ and $AAOD_{550}$ by $CTL_{ECHAM}$ compared to POLDER (see subsection

385 4.1).

Dust accumulation and coarse mode emissions increase everywhere, except over Iran and the Gobi desert for the coarse mode. Sea salt accumulation mode emissions are reduced almost everywhere in the world, while sea salt coarse mode emissions increase. This is a nice illustration on the importance of assimilating the $AE_{550-865}$ observations, since these

changes reduce the $AE_{550-865}$ overestimation compared to POLDER over the ocean. Organic carbon emissions increase



everywhere globally and approximately by a factor 3 in tropical Africa, 2.5 in the Amazon basin as well as Indonesia and by a factor 2 in Southeast Asia. Black carbon emissions increase approximately by a factor 3 in the United States, 1.5 in tropical Africa and are slightly reduced in Southeast Asia and parts of the Amazon basin. In all cases the underestimated $AAOD_{550}$ of the $CTL_{ECHAM}$ improves in $DAS_{ECHAM}$. Note that POLDER $AAOD_{550}$ is overestimated over several high-altitude areas (as

discussed in 4.1), thus emissions nearby to these areas may have been inflated since the correlation length of black carbon emissions perturbations in our data assimilation system is fairly big (25°). The $SO_2$ emissions increase in Europe as well as North America by about a factor 1.5 and remain almost the same over Southeast Asia. The same changes are observed for $SO_4$ total deposition.

Considering the relatively big changes in emissions, ranging from 1.5 to 3.0 for large portions of the globe, and the small improvements when evaluating the observables with all AERONET stations, it can be concluded that the network spatial coverage may not be sufficient to capture the global aerosol changes. This may be more relevant for $AE_{550-865}$ and $AAOD_{550}$ rather than $AOD_{550}$, where it is clearly improved (Figure 2). Note that $AE_{550-865}$ and $AAOD_{550}$ also improve against AERONET when we focus over specific areas (Figure 3 and Figure 4).

**4.3 Aerosol emission estimation from POLDER**

In this subsection the new global estimated emissions are presented and compared to previous studies. Some of these studies contain an ensemble of simulations (e.g. CMIP5, AEROCOM phase I and III), while others may include emissions based on data assimilation experiments. Note that the annual mean emissions for some studies may be regional and not global estimates (e.g. Chen et al., 2018; Escribano et al., 2017) and also may not refer to year 2006, which is the reference year for

our study. These issues, which are independent from inter-model differences in physics (e.g. emission schemes), chemistry parameterizations and prescribed emission inventories, may enhance the emissions differences from study to study. Thus, the comparison of our estimated emissions and the emissions from other studies is expected to differ and serves more as a qualitative comparison. The studies with an ensemble of models are presented in terms of the ensemble median, ensemble standard deviation and relative diversity, which is equal to the ratio of standard deviation to the median and then multiplied

by one hundred.

**4.3.1 Dust emissions**

Dust (DU) and Sea Salt (SS) global emission fluxes are shown in Figure 9. The emissions of these species are highly dependent on the simulated aerosol size range of each model, wind distribution in each model as well as the activation areas, where dust emissions are permitted, hence the emissions differ a lot from model to model (Wu, 2020). Previous studies have

also indicated that emissions fluxes for DU and SS are also highly resolution dependent (Guelle et al., 2001; Laurent et al., 2008). Specifically, ECHAM-HAM showed that DU emissions may differ by a factor of more than two globally, with local changes in emissions being even higher between a simulation at T63 ($CTL_{ECHAM}$) to T31 ($RES_{LOW}$) horizontal resolution,



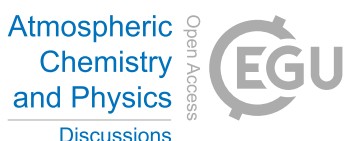

while smaller local differences were observed in SS emissions (FigureS 6). It is important to note here that the emissions estimation for a lower resolution (T31) data assimilation experiment (not shown) was very close (~1500 Tg·yr-1) to the
estimated emissions by the higher resolution (T63).

The global dust emissions of $CTL_{ECHAM}$ are 1105 Tg·year$^{-1}$ and are increased to 1419 Tg·yr$^{-1}$, a percentage change equal to 28%. These changes bring emissions closer to the estimates of many other studies, as indicated with the different coloured points in Figure 9. The ensemble median of AEROCOM I (including 14 models) is 1572 Tg·yr$^{-1}$, which lies quite close to
the estimates of this study.

As with the AEROCOM I models, AEROCOM III tends to underestimate AOD and overestimate AE over Sahara and middle east according to AERONET, which suggests that the coarse aerosol emissions are underestimated relative to the fine mode emissions (Gliß et al., 2021). The same can be seen in the $CTL_{ECHAM}$ and $DAS_{ECHAM}$ simulations (FigureS 7), with a
mean error of $AE_{550-865}$ at 0.055 and 0.146 respectively against AERONET. Note that POLDER $AE_{550-865}$ over Sahara is biased high based on AERONET, thus these observations were not assimilated (see Appendix A). The overestimated $AE_{550-865}$ suggests that the estimated dust emissions in $DAS_{ECHAM}$ should probably be higher, since the emissions of dust coarse mode, that correspond to the 98% of the total emitted dust globally, need to be higher.

The DU emissions ensemble median of CMIP5 models (15 models) is 2716 Tg·yr$^{-1}$ with a 2177 Tg·yr$^{-1}$ standard deviation and 80% diversity (C. Wu et al., 2020). Some of the factors that contribute to this diversity is the difference in the simulated size range (e.g. from 0.06μm to 63μm for some models and for <16μm for others), the global percentage where dust can be emitted that ranges from 2.9% to 18% and the differences in the spatial distribution of dust emissions.

The amount of the estimated dust emission due to data assimilation or observationally constrained methods in previous studies (Chen et al., 2018, 2019; Escribano et al., 2017; Huneeus et al., 2012; Schutgens et al., 2012) differs considerably both before and after observationally constraining the dust emissions for reasons that were already discussed. In all these studies dust emissions change between 27% to 62% with a median value of 46%. The percentage change of dust emissions due to the assimilated POLDER observations in the present study is 28%, that lies in the lower end of the percentage change
range of previous studies.

A recent study where dust emissions were constrained in terms of mass extinction efficiency, dust size distribution and dust optical depth revealed the importance of including the very coarse particles (up to 20μm in geometric diameter) for the total emitted dust mass in GCMs (Kok et al., 2021). According to the constrained experiment 1800 Tg·yr$^{-1}$ (with a 1 sigma
uncertainty between 1200 Tg·yr$^{-1}$ to 2700 Tg·yr$^{-1}$) were reported for emissions up to 10μm, which is close to our estimate and the ensemble of other studies. Contrary for emissions up to 20μm, 4700 Tg·yr$^{-1}$ (with a 1 sigma uncertainty between





3300 Tg·yr$^{-1}$ to 9000 Tg·yr$^{-1}$) were reported. The contribution of emitted particles between 10μm and 20μm to the total dust emissions was close to 65%, but the contribution to the total AOD$_{550}$ in the same size range was about 7%. The inclusion of dust coarse particles (>10μm) in GCMs is crucial for the total mass of dust emissions, absorption (Kok et al., 2021) and the nutrient contribution of dust to land and ocean ecosystems (Kim et al., 2014), but in terms of dust scattering the effect would be quite limited since their mass extinction efficiency relative to smaller particles is considerably smaller (particularly for the shortwave radiation).

### 4.3.2 Sea salt emissions

The SS emissions for the experiment CTL$_{ECHAM}$ is 1039 Tg·yr$^{-1}$, which in comparison with the other studies is considerably lower. The coarse mode, that contains 90% of the total emission mass of SS, is probably underestimated in the sea salt scheme that was used for our experiments (Long et al., 2011). This is supported also from an evaluation with POLDER, where the CTL$_{ECHAM}$ experiment overestimated AE$_{550-865}$ over the ocean (Figure 1e). The ensemble median of AEROCOM III is 4880 Tg·yr$^{-1}$ (excluding ECMWF-IFS) with a 1568 Tg·yr$^{-1}$ standard deviation and a 32% diversity (Gliß et al., 2021). ECMWF-IFS with an estimate of 50000 Tg·yr$^{-1}$ was not included since the emission scheme (Grythe et al., 2014) produces SS particles that are too large with very short lifetimes (Gliß et al., 2021). Note that the AEROCOM III ensemble median tends to underestimate the AE by 22%, mainly over the ocean, according to AATSR-SU observations, thus overestimating the SS particle size and in extent the mass flux of emissions (Gliß et al., 2021). (Textor et al., 2007) estimated based on a fraction of AEROCOM I models an ensemble median of 3830 Tg·yr$^{-1}$ with a 3830 Tg·yr$^{-1}$ standard deviation that results in a 100% diversity.

The assimilation of POLDER observations increases the global emissions to 1850 Tg·yr$^{-1}$ in DAS$_{ECHAM}$, which corresponds to a percentage change of +82% in respect to the CTL$_{ECHAM}$ experiment. Although SS emissions are still low (compared to the majority of AEROCOM III models for example), ECHAM-HAM can reproduce the AOD adequately both before and after the assimilation (Figure 1b and Figure 1c), indicating that the mass extinction coefficient (MEC) of the model is high. High MEC is related to more fine SS particles as the evaluation against POLDER AE$_{550-865}$ indicates (Figure 1). Further, a high MEC could be partially explained by the overestimated RH that enhances water uptake on SS and increases AOD. This topic is discussed further in subsection 4.4. Only one data assimilation study provides an aerosol emission estimate. (Schutgens et al., 2012) found that the SS emissions increased after assimilating AERONET stations and MODIS-Aqua AOD over ocean. It is noteworthy mentioning that their yearly emissions were estimated from a monthly (January 2009) experiment.

### 4.3.3 Organic aerosol emissions

In order to compare our emissions with other studies, OC emissions were converted to Organic Aerosol (OA) by multiplying with a factor 1.4 (Tegen et al., 2019). The OA global emission flux from the CTL$_{ECHAM}$ run is equal to 88.6 Tg·yr$^{-1}$.



AEROCOM III ensemble median is 116 Tg·yr$^{-1}$, with a large standard deviation (53 Tg·yr$^{-1}$) and diversity (46%). Inter-model differences between the AEROCOM III models are associated with differences on initial primary organic aerosols emissions, differences on secondary organic aerosol formation as well as differences in the conversion of OC from diverse sources of OA (Gliß et al., 2021). For example, the conversion factors to convert OC to OA can range between 1.4 to 2.6. These values are used by many AEROCOM III models that multiply all OC emissions, independently on the type of the source. But there are also models (e.g. NorESM2) that use different conversion factors depending on the source type, for example 1.6 for fossil fuel sources and 2.6 for biomass burning sources (Gliß et al., 2021).

The assimilated POLDER observations increase the OA emissions to 215.2 Tg·yr$^{-1}$ (+143%) in DAS$_{ECHAM}$, which is higher than any other emission estimation study. All previous data assimilation studies indicate an increase of OA emissions when observations are considered. The amount of increase differs from study to study, but the increasing signal is apparent in all, independently of the observations that were assimilated in each case. The emissions in Schutgens et al. (2012) and (Chen et al., 2019) are reported in OC, thus they were multiplied with 1.4 to get an approximation of OA emissions. The OA emissions increase in (Schutgens et al., 2012) from 116.2 Tg·yr$^{-1}$ to 190.4 Tg·yr$^{-1}$ (+64%), in (Huneeus et al., 2012) from 85 Tg·yr$^{-1}$ to 119 Tg·yr$^{-1}$ (+40%) and in (Chen et al., 2019) from 54.2 Tg·yr$^{-1}$ to 153.9 Tg·yr$^{-1}$ (+184%). Note that the increase of organic aerosol emissions in DAS$_{ECHAM}$ is significantly stronger for natural biomass burning sources (+193%) rather than anthropogenic sources (+115%).

### 4.3.4 Black carbon emissions

The Black Carbon (BC) global emission fluxes is 7.6 Tg·yr$^{-1}$ for the CTL$_{ECHAM}$ experiment. Since BC is highly absorbing, the estimated emissions will highly depend on the assimilation of SSA (or AAOD). Aerosol absorption information can be obtained by POLDER and as it has been shown previously the assimilation of absorbing observations are essential to correctly estimate BC mixing ratio and accurately simulate the absorption in a model (Tsikerdekis et al., 2021). The CTL$_{ECHAM}$ experiment underestimate AAOD$_{550}$ compared to POLDER, thus the BC emissions increase in DAS$_{ECHAM}$ to 13.3 Tg·yr$^{-1}$ (+75%). Previous data assimilation studies show similar increasing tendency as in OC emissions. Specifically, the BC emissions increase in (Huneeus et al., 2012) from 10 Tg·yr$^{-1}$ to 15 Tg·yr$^{-1}$ (+50%) and in Chen et al. (2019) from 6.9 Tg·yr$^{-1}$ to 18.4 Tg·yr$^{-1}$ (+166%).

Note that the biomass burning emissions of organic and black carbon are based on GFAS emissions. The biomass burning emissions in DAS$_{ECHAM}$ increase by 193% and 90% (not shown) respectively compared to CTL$_{ECHAM}$, which correspond to scaling factors equal to 2.93 and 1.90. These new scaling factors are distinctively estimated for organic and black carbon and are based on the assimilation of POLDER observations that includes absorption information, thus can be used from future studies to scale the GFAS emissions. Past studies have proposed a scaling factor of 3.4 for GFAS emissions based on an



AOD evaluation (Kaiser et al., 2012; Tegen et al., 2019; Veira et al., 2015), which was not applied in this study in order to estimate new scaling factors based on the assimilation of POLDER observations.

### 4.3.5 Sulfates and precursors emissions

The total deposition of $SO_4$, which we use as a proxy for $SO_4$ pseudo-emissions or rather the total chemical production of $SO_4$ in the atmosphere, along with the global emissions of $SO_2$ are depicted in Figure 9. The global pseudo-emissions of $SO_4$ are 210.9 Tg·yr$^{-1}$ for $CTL_{ECHAM}$. The pseudo-emissions of $SO_4$ for AEROCOM III ensemble median for the year 2010 is 143 Tg·yr$^{-1}$, with a 46.9 Tg·yr$^{-1}$ standard deviation and a 33% diversity (Gliß et al., 2021). ECHAM-HAM and ECHAM-SALSA have among the highest $SO_4$ pseudo-emissions in this ensemble (218 Tg·yr$^{-1}$ and 216 Tg·yr$^{-1}$ respectively), which indicates that the production of $SO_4$ from $SO_2$ is possibly higher or $SO_2$ loss is possibly lower in these two models compared to the other AEROCOM III models. Further, (Textor et al., 2007) noted that the differences in $SO_4$ production among AEROCOM I models remains almost the same, even when the same prescribed emissions of $SO_2$ are used. Thus, highlighting that inter-model differences in $SO_4$, may be caused primarily by differences in gain and loss processes rather than differences in the primary $SO_2$ emissions.

The $SO_2$ emissions of $CTL_{ECHAM}$ is 139.6 Tg·yr$^{-1}$. The respective value for the CEDS emission inventory used by the CMIP6 models is 123.4 Tg·yr$^{-1}$ (not shown in Figure 9). The $SO_2$ emissions in the HTAP v2 emission inventory for 2010 used in the (Chen et al., 2019) study is higher (175.6 Tg·yr$^{-1}$) than $CTL_{ECHAM}$, while $SO_2$ emissions in (Huneeus et al., 2012) for 2002 is closer (145.8 Tg·yr$^{-1}$) compared to $CTL_{ECHAM}$. Only the later study provides a new estimate for $SO_2$ emissions based on the assimilation of total and fine AOD of MODIS, that increased the $SO_2$ emissions to 165.8 Tg·yr$^{-1}$ (+14%). In $DAS_{ECHAM}$ the $SO_2$ emissions increase to 198.4 Tg·yr$^{-1}$ (+42%), which is higher than the reported emissions of (Chen et al., 2019) and Huneeus et al. (2012).

### 4.4 Overestimated relative humidity and the impact on aerosol optical properties

In this subsection we investigate the effect of errors in relative humidity, and resulting errors in aerosol water uptake, on the estimated emissions. In Figure 9, we compare the mean and standard deviation of the relative humidity profiles of ECHAM to ERA-5 and of the models of AEROCOM (8 models) and CMIP6 (7 models). ERA5 is set as the reference, since it is a reanalysis product where numerous meteorological observations were assimilated and compared to all the other GCMs the simulated RH should be closer to the truth. The majority of the models in this ensemble overestimate RH, both over land and ocean (Figure 10c,b), except the AEROCOM III simulation conducted with the models GFDL-AM4, GEOS and MIROC-SPRINTARS, where the simulated profile of relative humidity is closer to ERA5 and their horizontal resolution is at least two times higher compared to the other simulations. None of the models underestimate RH profile compared to ERA5. In addition to this ensemble, ECHAM-HAM simulations conducted for the year 2006 under different horizontal resolutions are also shown ($CTL_{ECHAM}$, $RES_{LOW}$). Clearly there is dependence between the horizontal resolution of ECHAM-HAM and its



capability to accurately simulate RH profiles. It is known that the current horizontal resolution of GCMs cannot directly resolve humidity's small scale spatial variability, thus it is parameterized (Lin, 2014). This is probably what is causing the differences in the RH profile compared to ERA5, but this is a topic that is out of the scope of our study. Note that the interannual ERA5 RH variability for the year 2006 (current study experiments) and 2010 (AEROCOM III) is miniscule (not shown).

The overestimation of RH for aerosol water uptake is most critical for the lower Troposphere (<~3km or about 700hPa), where RH is high enough (>50%) for water uptake to be relevant (Figure 11) and where most of the soluble aerosols exist. This overestimation is concentrated mostly over the ocean (FigureS 8), but there are also land areas where substantial overestimation of relative humidity is observed (e.g. East Asia). In order to quantify how aerosol properties are affected by the overestimation of RH profile by ECHAM-HAM, an additional experiment was conducted ($CTL_{ERA5}$) which is using the RH profile of ERA5 to determine the growth factor in ECHAM-HAM. Note that this modification affects only aerosol optical properties (scattering and absorption) in ECHAM-HAM, while the water cycle (precipitation and evaporation) of the model remains unaltered.

Figure 11 depicts the mean aerosol extinction profile for the experiments $CTL_{ECHAM}$ and $CTL_{ERA5}$. The aerosol extinction of insoluble particles is identical between the two experiments since they remain unaffected by aerosol water uptake changes. Contrary, the aerosol extinction of soluble particles in $CTL_{ERA5}$ exhibit considerably lower aerosol extinction compared to $CTL_{ECHAM}$. Over land this difference is maximum close to the surface and declines with height up to 600hPa (~3800m) where it becomes zero. Over the ocean, the difference is small close to the surface, peaks at 825hPa (~1500m) and slowly declines up to 650hpa (~3200m) where it becomes zero.

Note that over ocean ECHAM-HAM strongly overestimates RH profiles consistently over most grid cells, enhancing the growth of aerosols, that are mainly SS. Contrary over land RH is overestimated in East Asia, Europe and the eastern part of North America, where soluble $SO_4$ production is high and underestimated over Sahel and the western part of the America where insoluble DU particles are not affected by water uptake (FigureS 1). Consequently over ocean aerosol extinction profile differences (Figure 11c) matches the underestimation of RH by ECHAM-HAM (Figure 10c) while over land this is not the case (Figure 11b and Figure 10b). Most interestingly, over high density population areas (Eastern China, Europe, North America), where high emissions of anthropogenic $SO_2$ (precursor of $SO_4$) occur, the aerosol extinction difference between $CTL_{ECHAM}$ and $CTL_{ERA5}$ is even greater, indicating that aerosol extinction of anthropogenic induced aerosols is incorrectly inflated in ECHAM-HAM (and possibly in many other GCMs) because of the RH overestimation.

The global contribution of water $AOD_{550}$ to total $AOD_{550}$ changes from 62% to 52% from the experiment $CTL_{ECHAM}$ to $CTL_{ERA5}$. For reference, the contribution of water $AOD_{550}$ to total $AOD_{550}$ in (K. Zhang et al., 2012) was reported to be 70%



using ECHAM5-HAM2 which was nudged to ERA-40 for the year 2000. Although a 10% decrease is significant, water aerosol optical depth remains the largest contributor of total $AOD_{550}$ in $CTL_{ERA5}$, followed by DU (27%), SO4 (11%), OC (5%), SS (5%) and BC (1%).


Changes of $AOD_{550}$, $AE_{550-865}$ and $AAOD_{550}$ because of the overestimated RH are depicted in Figure 12. Globally $AOD_{550}$ is reduced by 0.015 (18%), $AE_{550-865}$ increases by 0.046 and $AAOD_{550}$ is virtually unchanged since BC and DU which contribute 96% of the global $AAOD_{550}$ (FigureS 3) are insoluble. Regionally, the $AOD_{550}$ change is by far strongest over East Asia, which can be explained by the presence of large loading of hydrophilic SO4 aerosol particles (FigureS 1e). The

same holds, to a lesser extent, for the eastern part of North America and Europe. Over ocean, largest AOD changes correspond to regions with high concentration of SS aerosols (FigureS 1b), within the Tropics and at high latitudes. $AE_{550-865}$ is affected by strong changes in the poles, where aerosol concentration is very low, so the global mean values are a bit misleading. $AE_{550-865}$ also increases over East Asia, eastern part of North America and Europe.

**4.4.1 Changes in emissions when considering the corrected relative humidity**

An additional data assimilation experiment was conducted using the relative humidity from ERA-5 (assumed to be the most accurate data available) to describe aerosol water uptake. The relative changes in aerosol emissions for this DA experiment ($DAS_{ERA5}$) compared to the standard DA experiment ($DAS_{ECHAM}$) is depicted in Figure 13. These changes are quantified by the ratio of $DAS_{ERA5}$ to $DAS_{ECHAM}$. The evaluation of aerosol optical properties of $DAS_{ECHAM}$ and $DAS_{ERA5}$ against POLDER and AERONET are very similar (not shown), suggesting that the emissions had to change differently in each

experiment in order to compensate the distinct differences in RH that affected aerosol optical properties.

As expected, strong changes occur for the soluble particles, SS and SO4. Overall, both the accumulation and the coarse mode emission of SS increase almost everywhere over the ocean. The increase in the accumulation mode is more pronounced in the South hemisphere. The considerable difference between the two DAS experiments is caused by the fact that in $DAS_{ERA5}$

aerosol particles are smaller (less water) and hence less efficient in extinction of light. So, more emission of more particles is needed to match the measured $AOD_{550}$ by POLDER. The global emissions in the $DAS_{ERA5}$ experiment are 2317 Tg·yr$^{-1}$.

As for SO4, $DAS_{ERA5}$ emissions distinctively increase over Southeast Asia by about 2 and to a lesser extent in Europe and North America (Figure 13 g and h). The results over Southeast Asia are particularly interesting since they could hint at a

potential big error in the bottom-up emissions inventories and/or could reveal underlying model errors with different signs that compensate each other. These changes are discussed in subsection 4.4.2 using additional evaluation with independent observations. The emission changes of the insoluble species (DU and BC) remain almost unchanged. Additionally, a very small reduction is observed for OC over North America and Southeast Asia, which barely reduces the $AOD_{550}$ by about 0.01.





### 4.4.2 Sulfates emissions in East Asia

In this subsection we are using additional observational datasets to evaluate the model over East Asia and further investigate the estimated emissions of $SO_4$ by $DAS_{ERA5}$ over Asia. Note that most of the $SO_2$ sources are in the eastern part of China (Figure 7h). The emissions of $SO_2$ and $SO_4$ for part of East Asia are depicted in Figure 14. Additional $SO_2$ estimates from bottom-up estimates are provided for comparison. DMS emissions are not shown since they contribute a very small fraction (about 3%) to the mass production of $SO_4$ and mostly over ocean.


The two CTL experiments and the $DAS_{ECHAM}$ are within the range of previous reported estimates, while the $SO_2$ and $SO_4$ emissions in $DAS_{ERA5}$ more than doubled compared to $CTL_{ERA5}$ (Figure 14). As already discussed, these large changes are caused by using more accurate relative humidity profiles for aerosol water uptake, that reduce $AOD_{550}$ significantly over the area and consequently the emission estimation system compensates for it by increasing $SO_2$ and $SO_4$ emissions. But since the

uncertainty of the bottom-up emission inventories is only 5.3% for eastern China (Chang et al., 2015), it is highly unlikely that $DAS_{ERA5}$ emissions are correct.

In Figure 15 and Figure 16 the experiments $CTL_{ERA5}$ and $DAS_{ERA5}$ are evaluated against various observations over eastern China. The mean difference of $AOD_{550}$ and $AE_{550-865}$ against POLDER improves from $CTL_{ERA5}$ to $DAS_{ERA5}$ (Figure 15). In

addition, the comparison of $AOD_{550}$ and $AE_{550-865}$ against AERONET improves from $CTL_{ERA5}$ to $DAS_{ERA5}$ (Figure 16). Note that the $AE_{550-865}$ for $CTL_{ERA5}$ in Figure 16h underestimates at low values and overestimates at large values which compensates for the mean error. The evaluation of surface $SO_4$ against CAWNET stations (values as reported in X. Zhang et al., 2012) did not provide conclusive evidence for improvement in the $DAS_{ERA5}$ experiment, since the mean error of $CTL_{ERA5}$ and $DAS_{ERA5}$ are of equal strength with a different sign (Figure 16 i-l).


Although aerosol optical properties are considerably better in $DAS_{ERA5}$, the evaluation of the experiments with OMI $SO_2$ column retrievals in Dobson units clearly indicates that $SO_2$ amount in the $DAS_{ERA5}$ is too high compared to OMI. This coincides with the bottom-up emission estimates discussed in Figure 14. According to these results we conclude that in $DAS_{ERA5}$ the $SO_2$ amount is overestimated but the $SO_4$ amount, which is the dominant aerosol type in this region, is

consistent with observations (both POLDER and AERONET) of $AOD_{550}$ and $AE_{550-865}$.

This inconsistency between $SO_2$ and $SO_4$ may be related to errors in the gain and loss mechanisms of $SO_4$, which also controls the atmospheric lifetime. Wet deposition is the dominant removal process for $SO_4$ globally and accounts for 97% of total deposition in ECHAM-HAM. On the other hand, wet deposition accounts only for 30% of the total deposition of $SO_2$.

Thus, biases in wet deposition will affect $SO_4$ lifetime more than $SO_2$. In ECHAM-HAM wet deposition and specifically below-cloud scavenging, simulates the removal rate of aerosol particles because of rain or snow depending on precipitation





rate, precipitation area and collection efficiency (K. Zhang et al., 2012). An evaluation with the Global Precipitation Climatology Project (GPCP) version 2.3 shows that both $CTL_{ERA5}$ and $DAS_{ERA5}$ overestimate precipitation by more than 50% over the eastern China domain (Figure 15). This overestimation should decrease the modelled atmospheric lifetime of
$SO_4$ and lower the AOD in the area. In order to match observed AOD values, this is compensated in $DAS_{ERA5}$ by too high $SO_2$ and $SO_4$ emissions.

Globally the total mass production of $SO_4$ particles in ECHAM-HAM is mainly driven by oxidation of dissolved $SO_2$ in-clouds by $O_3$ and $H_2O_2$ (72.5%), followed by an oxidation reaction of OH with $SO_2$ (20.9%) and OH with DMS (3.3%) in
cloud free conditions. Finally, a small percentage is contributed by direct emissions of aerosol $SO_4$ (2.5%). Based on MODIS-Terra the cloud Liquid Water Path (LWP) over eastern China is overestimated by more than 50% in both $CTL_{ERA5}$ and $DAS_{ERA5}$, which potentially accelerates the in-cloud production of $SO_2$ to $SO_4$ in ECHAM-HAM and inflates the AOD in the area. In an inverse emission estimation like $DAS_{ERA5}$, this would lead to a reduction in $SO_2$ and $SO_4$ emissions. The fact that the SO2 emissions increase to unrealistically large values suggests that errors caused by too strong wet deposition
dominates over the error caused by too much $SO_4$ in-cloud production. A future study with additional sensitivity studies may fully disentangle and quantify the biases of these processes.

Additional causes for the underestimated $AOD_{550}$ in $CTL_{ERA5}$, that lead to an excessive increase of $SO_2$ emissions in $DAS_{ERA5}$, may be the lack of ammonium nitrate ($NH_4NO_3$) in ECHAM-HAM. Particulate nitrates (hereafter refer to as
nitrates) forms either through aqueous chemical reaction between gaseous ammonia ($NH_3$) and gaseous nitric acid ($HNO_3$) or with heterogeneous reaction of nitrogen species (e.g. $HNO_3$, $NO_3$ and $N_2O_5$) on the surface of dust and sea salt particles (Bian et al., 2017). Some of the AEROCOM III models that simulate both nitrates and sulfates report that the global mean $AOD_{550}$ of sulfates (0.0392) is five times greater than the respective global mean $AOD_{550}$ of nitrates (0.0072) (Bian et al., 2017). Further, the global contribution of nitrates $AOD_{550}$ to the global total $AOD_{550}$ according to the ensemble of all
AEROCOM III models is about 2% to 3% (Gliß et al., 2021). Although globally the effect of nitrates $AOD_{550}$ is limited, locally over agricultural highly polluted areas can be considerably higher. According to (Park et al., 2014) the Nitrate $AOD_{550}$ for the year 2006 accounts for more than 15% of the total $AOD_{550}$ over the East Asia domain and about 20% at AERONET sites over the same domain. The AERONET $AOD_{550}$ for a similar domain used in Park et al. (2014) is 0.539 (Figure 16a), from which 0.108 (20% of 0.539) is contributed by nitrates. Consequently, ECHAM-HAM underestimates
$AOD_{550}$ by about 0.10 because it does not consider nitrate aerosol.

The missing $AOD_{550}$ over East Asia could also be explained if the water uptake process is underestimated in ECHAM-HAM, i.e. if the growth factors at given relative humidity are underestimated. However, the results in (Burgos et al., 2020) do not suggest that, because they showed that the models ATRAS, CAMS and CAM-OSLO, that use the κ-Köhler parameterization
for aerosol water uptake with very similar kappa values for all aerosol species as ECHAM-HAM, have a good agreement in

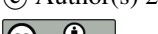



scattering enhancement factors with 22 different sites (see Burgos et al. 2020 for more details), though with a small positive bias. Thus, the errors in scattering enhancement due to water uptake in ECHAM-HAM is not underestimated and cannot be the cause of the low $AOD_{550}$ in $CTL_{ERA5}$.

**5 Conclusions**

We have estimated aerosol emissions for the year 2006 based on the assimilation of POLDER observations related to the aerosol amount, size, and absorption ($AOD_{550}$, $AE_{550-865}$ and $SSA_{550}$). The data assimilation system was developed using an existing ensemble Kalman smoother code (Schutgens et al., 2012) that was modified for the model ECHAM-HAM (Tsikerdekis et al., 2022). The global aerosol emissions of all species increase compared to the prior emissions from bottom-up inventories after the assimilation of POLDER observations, specifically 28% for dust, 75% for sea salt, 143% for organic

carbon, 75% for black carbon and 39% for sulfates. Specifically, the biomass burning emissions of organic aerosol and black carbon increase by 193% and 90% respectively. The changes lead to a simulated aerosol state that is overall in a better agreement with the assimilated (POLDER) and independent (AERONET and MODIS) observations. However, we found that the global spatial distribution of the AERONET stations cannot fully capture the changes of observables due to the adjusted emissions.


The a-priori and estimated emissions are compared with the reported emissions used in the AEROCOM and CMIP5 ensemble of models, as well as other observationally constrained studies. The new dust emissions are very close to the ensemble median of AEROCOM, and match quite well the estimated emissions reported by other data assimilation studies (Hueneeus et al., 2012; Kok et al., 2021). New sea salt emissions are close, but still are on the lower end, compared to the

emissions from other studies. A possible explanation is that the ECHAM-HAM sea salt scheme we use (Long et al., 2011) underestimates the coarse sea salt particles, which is characterized by short lifetime and small contribution to $AOD_{550}$ but has a high impact on total emissions mass.

The derived organic aerosol emissions are higher than the upper bound of the AEROCOM range, as well as higher than any

other top-down estimates. There are four top-down emission estimates (including the present one) and all of them lead to a significant increase compared to the (bottom up) prior emission (Schutgens et al., 2012; Chen et al., 2019; Huneeus et al., 2012). However, the 4 different estimates span a considerable range and the estimate of the present work yields the highest emission for organic aerosol. The derived black carbon emissions in this study are closer to the estimated emissions by Chen et al. (2019) as well as Huneeus et al. (2012) and all agree that the emissions should be higher than bottom up estimates.


In this study we estimate emissions of OC and BC separately for either biomass burning or other sources combined. Based on the data assimilation changes we observe in the prior GFAS emissions we propose scaling factors equal to 2.93 and 1.90





for OC and BC respectively. Past studies have proposed a scaling factor of 3.4 for GFAS emissions based on AOD (Kaiser et al., 2012; Veira et al., 2015; Tegen et al., 2019). These new scaling factors are based on the assimilation of POLDER observations that include absorption information, could be adopted to future studies to scale the GFAS emissions.

We found that estimated sulfates emissions are very sensitive to the relative humidity profile (because of hygroscopic growth), and that ECHAM-HAM significantly overestimates relative humidity. The same holds for virtually all AEROCOM and CMIP6 models. When the aerosol water uptake process in ECHAM-HAM uses the relative humidity of ERA5, the global $AOD_{550}$ reduces by 0.015, while the reduction over East Asia can be higher than 0.2. This can be explained by smaller wet-growth of aerosols due to lower relative humidity. Thus, we conducted a second yearly data assimilation experiment where new emissions were estimated when the aerosol wet growth in the model uses ERA5 RH (instead of ECHAM-HAM RH). The global emissions of sulfates increased by 85%, which is considerably higher than the increase in the base experiments. For the same reason, sea salt emissions increased by 123%. As expected, the emissions of insoluble (dust, black carbon) or not very soluble (organic carbon) species were much less sensitive to the relative humidity.

Specifically, over East Asia, the new emissions of sulfur dioxide (primary precursor for sulfates) more than doubles in the new set-up with ERA5 relative humidity. The new estimates are considerably higher than all the bottom-up emission inventories. A thorough evaluation with independent observations over East Asia reveal that the lack of $AOD_{550}$ that leads to an intense increase of sulfur dioxide emissions is possibly caused by (i) the overestimated precipitation that enhances wet deposition and reduces the aerosol lifetime and AOD550 (ii) or the missing nitrates on ECHAM-HAM that may contribute by up to 15% of AOD (Park et al., 2014). Conversely, a compensating effect of overestimated cloud liquid water path, that enhances the in-cloud production of $SO_4$ particles, was also found over the same area, but considering the lack of $AOD_{550}$ this effect is likely less important. A future study should study in more detail the gain (e.g. conversion speed of $SO_2$ to $SO_4$) and loss (dry and wet deposition) processes in the model.

**Appendix A**

The uncertainty of POLDER observations is estimated by evaluating it with AERONET for different AOD bins. FigureA 1 depicts the uncertainty for AOD, AE and SSA. Lines illustrate the uncertainty (left axis) and bars the number of paired POLDER and AERONET observations that were used in each $AOD_{550}$ bin to estimate the uncertainty (right axis). $AOD_{550}$ relative uncertainty is lower than 50% for POLDER $AOD_{550}$ greater than 0.1 and it steadily decreases both over land and ocean as POLDER $AOD_{550}$ increases. The land and ocean retrievals are notably different for $AE_{550-865}$, where the mean difference in uncertainty for all $AOD_{550}$ bins is 0.466. Thus, it is expected that the over land $AE_{550-865}$ will have little to no effect when assimilated, compared to the over ocean $AE_{550-865}$. Further, we found that over Sahara $AE_{550-865}$ is overestimated by POLDER by 0.524, thus these observations were not used in the assimilation. The uncertainty over land $SSA_{550}$ is higher

than 0.05 for $AOD_{550}$ bins lower than 0.4 and decreases (between 0.04 to 0.02) for $AOD_{550}$ higher than 0.4. Which strongly suggests that for high polluted areas, absorption is retrieved by POLDER with reasonable accuracy. The over ocean $SSA_{550}$ uncertainty was estimated only up to 0.4 $AOD_{550}$ bin due to the lack of AERONET observations for higher AODs. Currently a new version of POLDER SRON retrievals is being prepared, which is expected to yield a significantly improved POLDER aerosol product.


In addition to the uncertainty of observations presented in FigureA 1, a representation error was added to the uncertainty of $AOD_{550}$ and $AE_{550-865}$ observations. Specifically, an analysis was performed using CAMS reanalysis in two resolutions, one in 1° x 1° (resolution of POLDER level-3) and one in coarser resolution 1.875° x 1.875° (resolution of ECHAM-HAM). The objective of this analysis was to determine how well an observation on a 1° x 1° horizontal resolution represents the

respective observations on a 1.875° x 1.875° model resolution. This was done by firstly collocating the data of the two resolutions. Obviously, each coarse resolution paired with multiple high resolution observations. For each paired observation the differences were calculated. Then the standard deviation of the differences for each 1.875° x 1.875° grid box was estimated. The global mean value of all standard deviations was used as a representation error, distinctively for the $AOD_{550}$ and $AE_{550-865}$ case. The added representation error for $AOD_{550}$ is 0.022 and for $AE_{550-865}$ is 0.062. The respective values for a

coarser resolution (3.75° x 3.75°) are 0.045 and 0.120 for $AOD_{550}$ and $AE_{550-865}$ respectively. No representation error was used for the observations of $SSA_{550}$, since the SSA was not available in the Atmosphere Monitoring Service (ADS) for the CAMS reanalysis.

**Appendix B**

The prior emission uncertainties are based on an ensemble of simulations where in each member the emissions of each

aerosol species have been distinctively perturbed. We multiplied the emissions of each ensemble member by sampling numbers from a positive skewed distribution with a distinctive standard deviation for each species and a mean of one. The distinctive standard deviations were based on the standard deviation of the differences between ECHAM-HAM minus POLDER daily AOD. We assumed that the standard deviation of these differences filtered over specific locations can be used as a proxy for emissions uncertainty by species.


In FigureA 2 the estimated emissions uncertainty (standard deviation differences of AOD, explained above) is depicted as a function of several emissions percentiles, where a low percentage contains all the daily grid-box emissions and high percentage contains only the highest daily grid-box emissions. Theoretically, when the emissions threshold is high the contribution of that specific aerosol species to the total aerosol load in the atmosphere increases, thus the emission

uncertainty will be more representative of that species.





The current analysis gives little information on the emissions uncertainty over low emission sources, thus we assume that low and high emission sources share the same uncertainty. The emissions uncertainty for this study was based on the median (50%) emissions threshold, in order to filter out cases where multiple aerosol species are mixed in the atmosphere but also include sources with relative mediocre strength. Note that this approach attributes all modeling errors that may affect aerosol optical properties (e.g. transport, deposition, water uptake, aerosol chemical production) as emissions uncertainty. Consequently, the emissions uncertainty is possibly overestimated in some cases. For example, previous study suggests that fossil fuel emissions are lower than 20% for BC and lower than 42% for SO2 (Granier et al., 2011). Further, note that since we are using AOD as a proxy for emissions uncertainty the absorbing aerosols (BC) will have similar uncertainty with the scattering aerosol species (OC) that are emitted in the same locations.



**Data availability.** The model simulations and the assimilated POLDER data are available from Zenodo at the following link: https://doi.org/10.5281/zenodo.7565093. The ECHAM-HAM version that was used in this study can be found in the following repository: https://redmine.hammoz.ethz.ch/projects/hammoz/repository/1/show/echam6-hammoz/branches/uni_amsterdam_vrije/WC20220422 (last access: 17 January 2023). This repository can be accessed after
registration at https://redmine.hammoz.ethz.ch/projects/hammoz (Hammoz, 2023). ERA-interim and ERA-5 data are freely available from https://doi.org/10.24381/cds.bd0915c6 (Hersbach et al., 2018) after registration. The AERONET (https://aeronet.gsfc.nasa.gov/) data are freely available.

**Supplement.** The supplement related to this article is available online at: …

**Author contributions.** AT organized the experiments with the help from OPH and NAJS. AT conducted the experiments
and performed the analysis. QZ acquired the AEROCOM III data. AT prepared the manuscript with contributions from all co-authors.

**Competing interests.** The contact author has declared that neither they nor their co-authors have any competing interests.

**Disclaimer.** Publisher's note: Copernicus Publications remains neutral with regard to jurisdictional claims in published maps and institutional affiliations.

**Acknowledgements.** We thank the principal investigators, co-investigators and their staff for establishing and maintaining the AERONET sites used in this investigation. This work was carried out on the Dutch national e-infrastructure with the support of SURF Cooperative.

**Financial support.** This research was funded by NWO/NSO project "AEROSOURCE: Estimation of Aerosol Emissions from Polarization Data" (grant no. ALWGO.2017.008).

**Review statement.** …

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

**Table 1. Emission types that are distinctively perturbed and estimated (state vector) by the assimilation system. Fossil fuel refers to all emissions except biomass burning, which to a large extent includes mainly fossil fuel emissions but also other natural emissions like biogenic emissions. Biomass burning emissions include both natural and anthropogenic induced fires. SO2, DMS and SO4**
**share the same perturbations distinctively for biomass burning and fossil fuel. The sulfates in the atmosphere are mainly produced by emitted SO2, followed by DMS. Direct emissions of SO4 are modeled as 2.5% of the SO2 emissions.**

| Species | Mode | Hygroscopicity | Sector |
|---|---|---|---|
| DU | Accumulation | Insoluble | - |
| DU | Coarse | Insoluble | - |
| SS | Accumulation | Soluble | - |
| OC | Aitken | Insoluble | Biomass Burning |
| OC | Accumulation | Soluble | Biomass Burning |
| OC | Aitken | Insoluble | Biomass Burning |
| OC | Aitken + Accumulation | Insoluble | Fossil Fuel |
| BC | Aitken | Insoluble | Biomass Burning |
| BC | Aitken | Insoluble | Fossil Fuel |
| $SO_2$ / DMS / $SO_4$ | Aitken + Accumulation + Coarse | Soluble | Biomass Burning |
| $SO_2$ / DMS / $SO_4$ | Aitken + Accumulation + Coarse | Soluble | Fossil Fuel |

**Table 2. Experiments overview.**

| Experiment | Assimilation | Resolution | RH for water uptake |
|---|---|---|---|
| $CTL_{ECHAM}$ | - | 1.875° x 1.875° | ECHAM-HAM |
| $DAS_{ECHAM}$ | POLDER AOD, AE, SSA | 1.875° x 1.875° | ECHAM-HAM |
| $CTL_{ERA5}$ | - | 1.875° x 1.875° | ERA5 |
| $DAS_{ERA5}$ | POLDER AOD, AE, SSA | 1.875° x 1.875° | ERA5 |
| $RES_{LOW}$ | - | 3.75° x 3.75° | ECHAM-HAM |



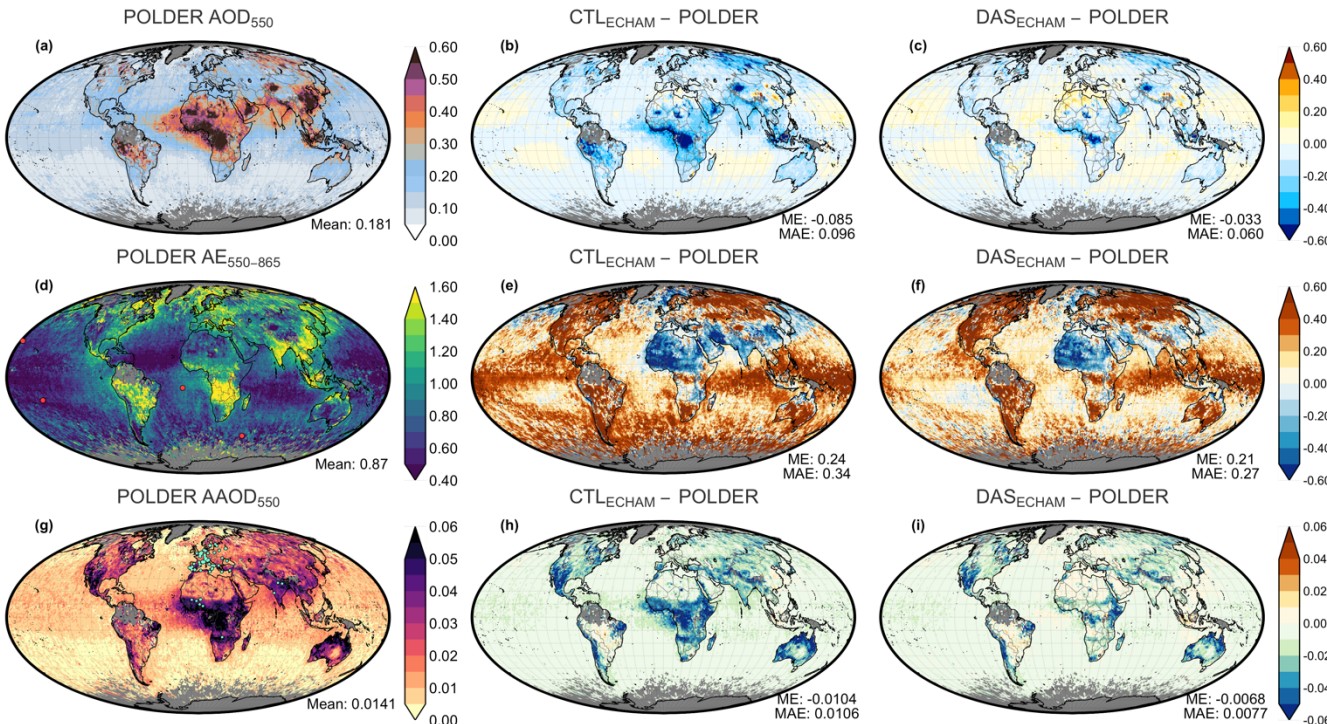

**Figure 1.** An evaluation of $CTL_{ECHAM}$ and $DAS_{ECHAM}$ experiments, based on POLDER for the year 2006. First column depicts POLDER (a) $AOD_{550}$, (b) $AE_{550-865}$ and (c) $AAOD_{550}$, while the second and the third column displays the differences $CTL_{ECHAM}$ − POLDER and $DAS_{ECHAM}$ − POLDER respectively. The global mean, the global mean error (ME) and the global mean absolute error (MAE) is depicted at the right bottom corner of each plot. The points on the d and g depict AERONET stations used for the plots of Figure 3 and Figure 4 respectively.







**Figure 2. An evaluation of POLDER (first column), CTL$_{ECHAM}$ (second column) and DAS$_{ECHAM}$ (third column) based on AERONET for the year 2006. The first, second and third row correspond to the variables AOD$_{550}$, AE$_{550-865}$ and AAOD$_{550}$ respectively. The OBS mean refers to AERONET in all plots. The Mean Error (ME), Mean Absolute Error (MAE), Root Mean**
**Square Error (RMSE), Pearson Correlation (R) and the number of data points used in each case (N) is depicted at the top-left of each subplot. The AOD$_{550}$ and AE$_{550-865}$ evaluation is based on AERONET Version 3 Direct Sun Algorithm Level 2.0, while the AAOD$_{550}$ evaluation is based on AERONET Version 3 Direct Sun and Inversion Algorithm Level 1.5.**







**Figure 3.** AE$_{550-865}$ evaluation of CTL$_{ECHAM}$ and DAS$_{ECHAM}$ based on selected AERONET stations (red points in Figure 2d) for the year 2006. These stations are in isolated islands over the ocean in order to capture the changes of AE$_{550-865}$ due to SS emission changes. The shaded areas depict the 2D density estimate scaled to a maximum of one for 0.3, 0.6 and 0.9 intervals. The Mean Error (ME), Mean Absolute Error (MAE), Pearson Correlation (R) and the number of data points used in each case (N) is depicted for each subplot. The evaluation is based on AERONET Version 3 Direct Sun Algorithm Level 2.0.





Figure 4. AAOD550 evaluation of CTL$_{ECHAM}$ and DAS$_{ECHAM}$ based on selected AERONET sites (cyan points in Figure 2g) for the year 2006. These stations are selected over regions where natural and anthropogenic emissions of BC occur. The shaded areas depict the 2D density estimate scaled to a maximum of one for 0.3, 0.6 and 0.9 intervals. The Mean Error (ME), Mean Absolute Error (MAE), Pearson Correlation (R) and the number of data points used in each case (N) is depicted for each subplot. The evaluation is based on AERONET Version 3 Direct Sun and Inversion Algorithm Level 1.5.





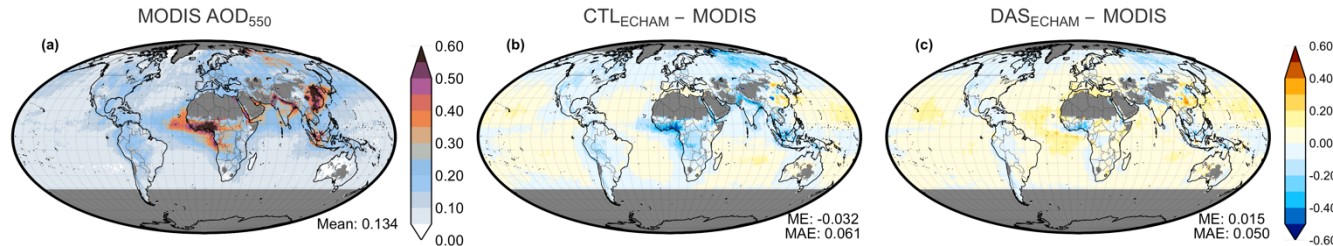


**Figure 5.** An evaluation of CTL$_{\text{ECHAM}}$ and DAS$_{\text{ECHAM}}$ experiments, based on MODIS for the year 2006. First column depicts MODIS AOD$_{550}$, while the second and the third column displays the differences CTL$_{\text{ECHAM}}$ − POLDER and DAS$_{\text{ECHAM}}$ − POLDER respectively. The global mean, the global mean error (ME) and the global mean absolute error (MAE) is depicted at the right bottom corner of each plot.

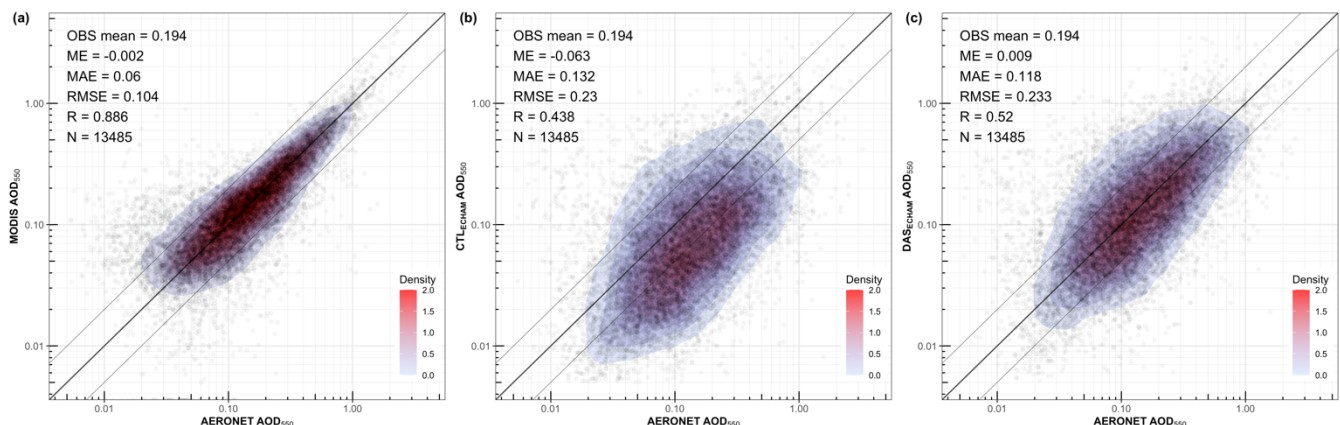


**Figure 6.** An AOD$_{550}$ evaluation of MODIS (first column), CTL$_{\text{ECHAM}}$ (second column) and DAS$_{\text{ECHAM}}$ (third column) based on AERONET for the year 2006. The OBS mean refers to AERONET in all subplots. The Mean Error (ME), Mean Absolute Error (MAE), Root Mean Square Error (RMSE), Pearson Correlation (R) and the number of data points used in each case (N) is depicted at the top-left of each subplot. The AOD$_{550}$ evaluation is based on AERONET Version 3 Direct Sun Algorithm Level 2.0.






**Figure 7. Aerosol emissions (kg km⁻² day⁻¹) of CTL$_{ECHAM}$ experiment for 2006. The global mean is depicted at the right bottom corner of each map. The pseudo-emissions of SO$_4$ are based on SO$_4$ total deposition.**




**Figure 8. Relative changes of aerosol emissions due to the assimilated POLDER observations (DAS$_{ECHAM}$ divided to CTL$_{ECHAM}$) for 2006. The global mean is depicted at the right bottom corner of each map. Gray grid cells contain emissions lower than the global median value of each species and are excluded from these maps.**






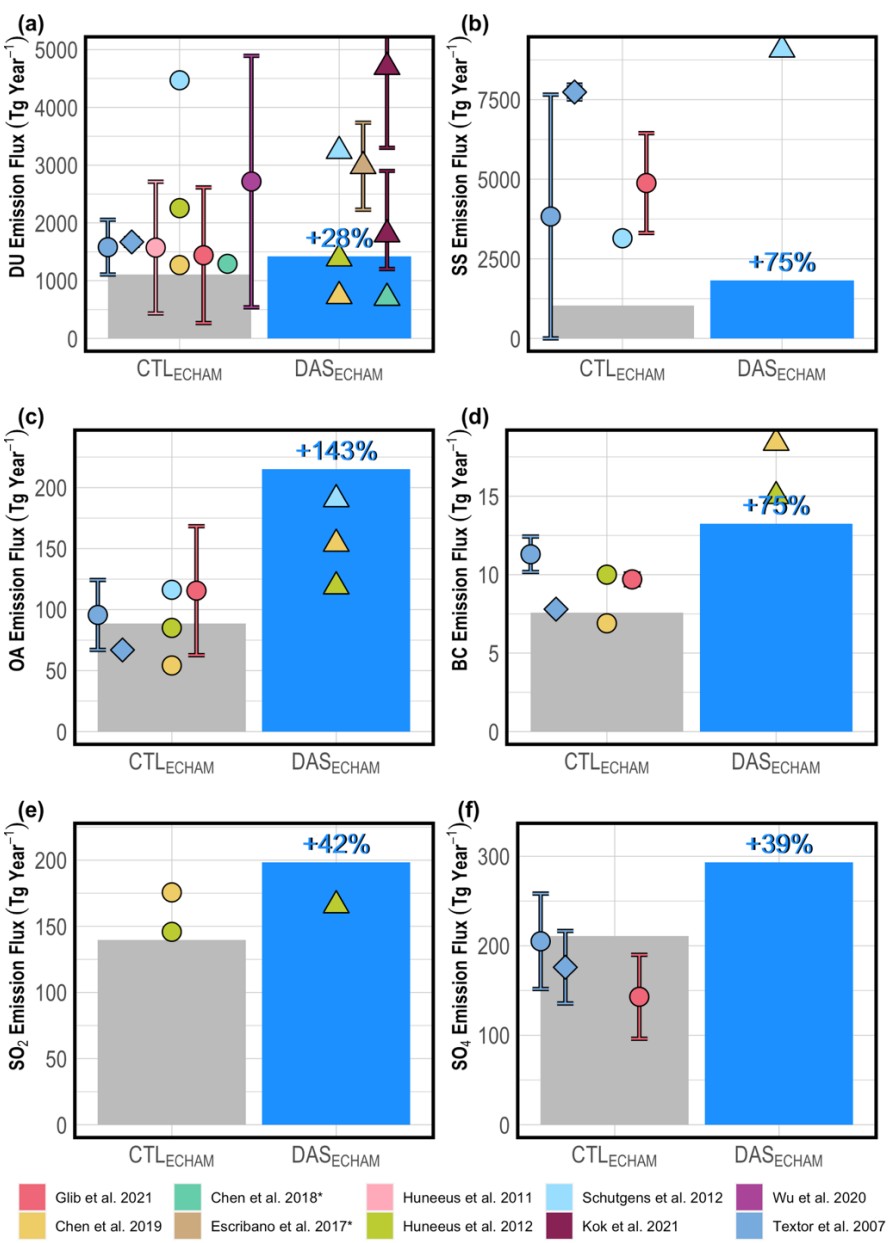

**Figure 9. Global aerosol emissions of 2006 for Dust (DU), Sea Salt (SS), Organic Aerosol (OA), Black Carbon (BC), Sulfur Dioxide (SO2) and total deposition of Sulfates (SO₄) (Tg yr⁻¹). The percentage change of the estimated emissions over DAS_ECHAM is estimated based on the emissions of CTL_ECHAM respectively. Circles depict the reported emissions from other studies. Diamond depicts the sensitivity study in Textor et al. (2007) which is explained in the text. Triangles illustrate the emissions estimated from past data assimilation studies. OA is estimated by multiplying the emissions of OC with 1.4 for the experiments of this study, as well for the reported emissions in Schutgens et al. (2012) and Chen et al. (2019). SO₄ total deposition is used as a proxy for SO₄ pseudo-emissions. SO₂ emissions for Chen et al. (2019) and Huneeus et al. (2012) were reported in Tg S yr⁻¹, thus they were multiplied with 2 in order to be converted in Tg SO₂ yr⁻¹. The asterisk symbol on some studies indicate that the emissions reported are regional and not global. The yearly emissions from Schutgens et al. (2012) are an extrapolation of a single month's (January) experiment. The two Kok et al. (2021) estimates refer to emissions for DU particles up to 10μm (low estimate) and up to 20μm (high estimate) in geometric diameter (see text for more details).**





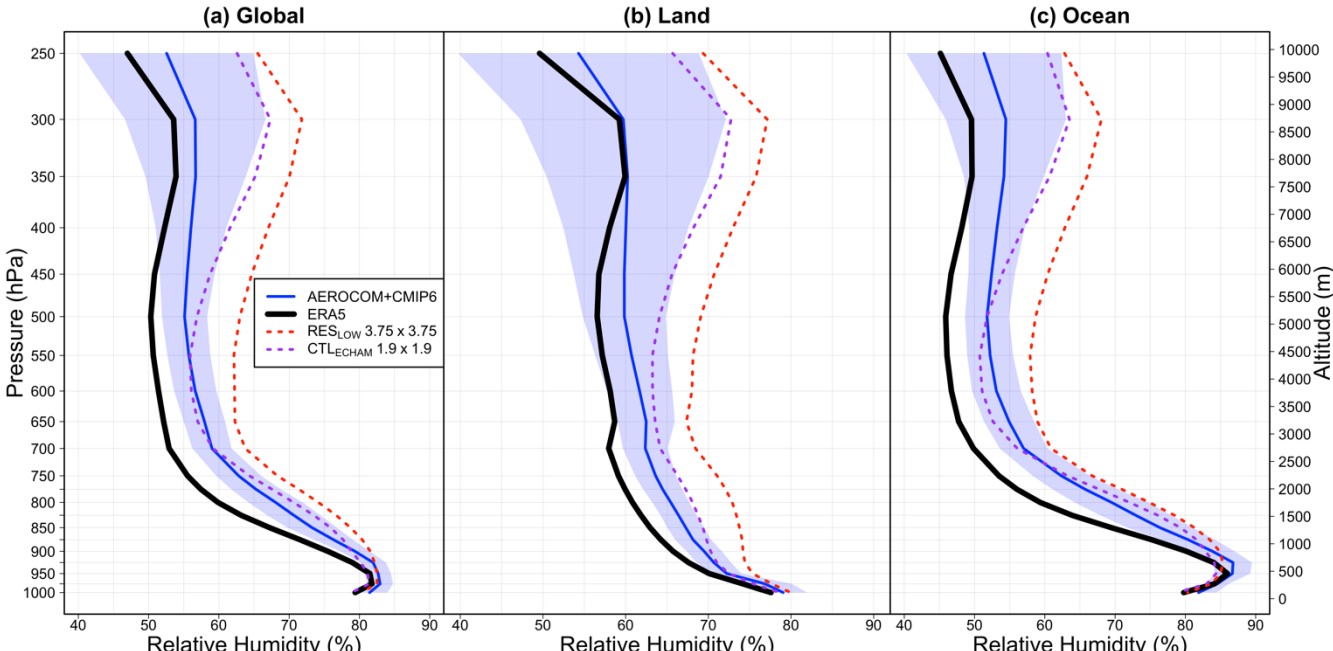

**Figure 10. Relative humidity profile for a multi-model ensemble mean from 15 simulations that includes AEROCOM III and CMIP6 models (blue) along with the ERA5 (black) for the year 2010. The shaded area represents the standard deviation of the ensemble. The experiments CTLECHAM and RESLOW are also depicted.**

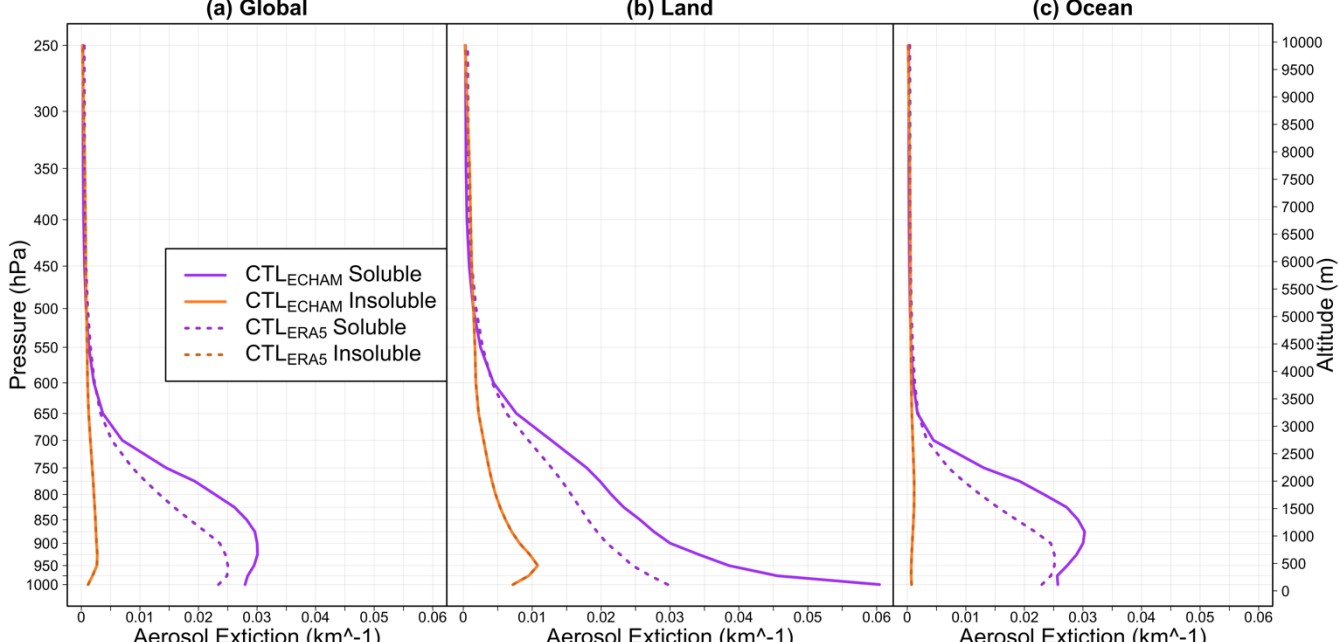

**Figure 11. Aerosol extinction profile (km$^{-1}$) of CTL$_{ECHAM}$ and CTL$_{ERA5}$ for soluble and insoluble aerosols.**





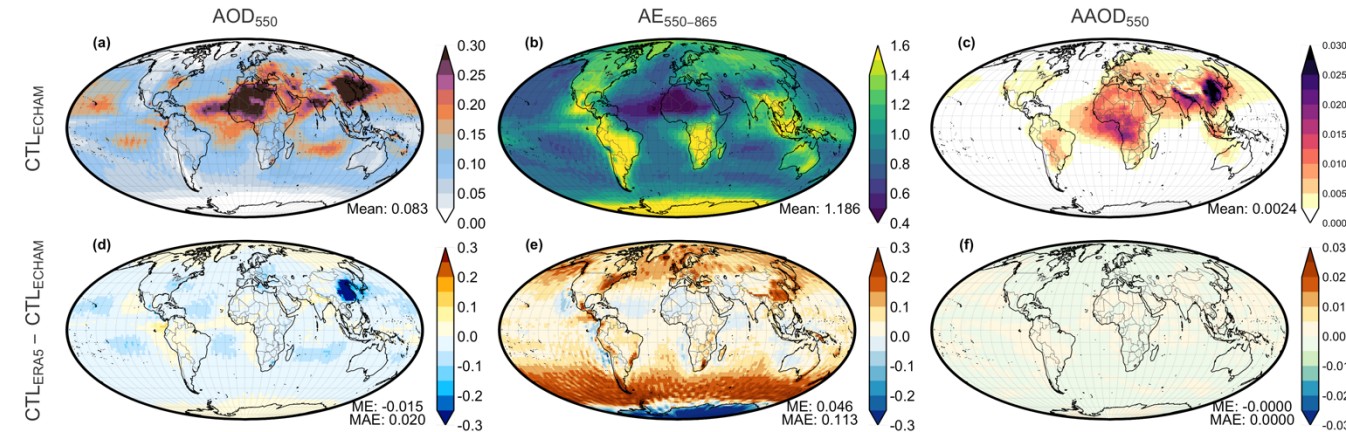

**Figure 12. Aerosol optical properties of CTL_ECHAM and the differences between CTL_ERA5 and CTL_ECHAM.**





**Figure 13. Relative changes of aerosol emissions after accounting for the correct relative humidity for aerosol water uptake (DAS_{ERA5} divided to DAS_{ECHAM}) for 2006. The global mean is depicted at the right bottom corner of each map. Grey grid cells contain emissions lower than the global median value of each species and are excluded from these maps.**





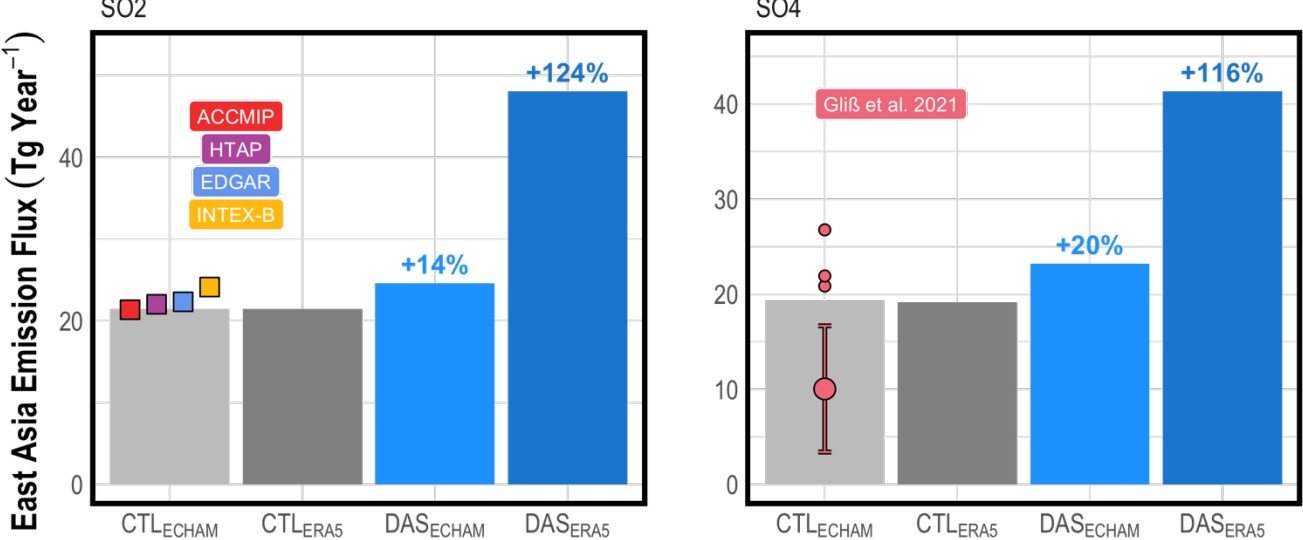

**Figure 14. Aerosol emissions over China for SO$_2$ and SO$_4$ (Tg yr$^{-1}$). The percentage change of the estimated emissions over DAS$_{ECHAM}$ and DAS$_{ERA5}$ is estimated based on the emissions of CTL$_{ECHAM}$ and CTL$_{ERA5}$ respectively. Bars show the sum of the emissions for eastern China (100° to 120° E, 24° to 44° N). The squares depict the annual emissions of 2006 for four bottom up inventories (ACCMIP, HTAP, EDGAR and INTEX-B) over the same domain as reported on Chang et al. (2015).**

1105





**Figure 15.** The (a) POLDER AOD$_{550}$, (b) POLDER AE$_{550-865}$, (c) OMI SO$_2$ in Dobson units, (d) GPCP Precipitation and (e) MODIS-Terra cloud liquid water over eastern China. The second and third column shows differences CTL$_{ERA5}$ − observations and DAS$_{ERA5}$ − observations respectively.





**Figure 16. An evaluation of the CTL$_{ERA5}$ and DAS$_{ERA5}$ experiments for AOD$_{550}$ and AE$_{550-865}$ against AERONET (subplot a to h). In the maps the inner circle depicts the mean AE$_{550-865}$ of all AERONET stations within a grid cell of the model while the outer circle depicts the difference between experiments minus AERONET. The size of the points is analogous to the number of the available data points in each case. The scatterplots use all the available data points of the displayed stations. An evaluation of the same two experiments for SO4 surface concentrations against CAWNET (as reported in Zhang et al. (2012b) for 2006 and 2007) is shown in the subplots i-l.**



**FigureA 1. POLDER uncertainty for AOD550, AE550-865 and SSA550 based on an AERONET evaluation for several POLDER AOD550 bins. Red and blue lines depict the uncertainty of over land and over ocean retrievals respectively (left axis). The respective colored bars illustrate the number of collocated POLDER and AERONET retrievals were used to calculate the observables uncertainty in each AOD550 bin (right axis) and N depicts the total number. Note AOD550 uncertainty was estimated in relative terms, by dividing with AERONET AOD550. The AOD550 and AE550-865 evaluation is based on AERONET Version 3 Direct Sun Algorithm Level 2.0, while the AAOD550 and SSA550 evaluation is based on AERONET Version 3 Direct Sun and Inversion Algorithm Level 1.5.**

1120







l125

**FigureB 1. Emission uncertainty as a function of emission threshold for each parameter. Emission uncertainty (yy' axis) is set as the standard deviation of daily AOD550 differences of POLDER – ECHAM-HAM for the year 2006. The emission threshold (xx' axis) depicts the percentile of daily emissions. The SO4 emission uncertainty represents also the emission uncertainty used for SO2 and DMS. Note that for DU, SS and OC multiple modes are perturbed distinctively, but the modes of those species use the same emission uncertainty. The yellow shaded area highlights the emission uncertainty used in this study, where the emission threshold is set at 50% (includes sources with higher value than the median). For more details see text in Appendix B.**

l130