# Peer review of "Assimilation of POLDER observations to estimate aerosol emissions"

_Atmospheric Chemistry and Physics, 2023_

## Author Comment (AC1)

**Response to Referee #1 for the manuscript: "Assimilation of POLDER observations to estimate aerosol emissions"**

Dear Referee #1,

Thank you for reviewing our manuscript. Your comments help us to improve and define better some aspects of our work. Below you can find our point-by-point responses to all of your comments.

Best regards,
On behalf of all co-authors
Athanasios Tsikerdekis

**Format**
Questions
Responses
"Quotes from the manuscript and revised or added text."

**Comments**

What I missed is some information for the meteorological set up of the simulations with ECHAM-HAM climate model (e.g. if the simulations are nudged, spin-up time of the simulations, if is there is a specific reason for the selection of the year 2006).

Thank you for your suggestion, indeed this information was missing from the document. We added a supplement TableS 1 (shown below), along with all the references in the main manuscript. The TableS 1 is referred in the subsection 3.2 Experimental Setup: "A list of selected meteorological and aerosol options used for the experiments is presented in TableS 1."

In addition, the year 2006 was selected based on the availability of POLDER SRON observations. A sentence was added in subsection 2.1: "In the present study aggregated ($1° \times 1°$) POLDER data are used in the assimilation for the year 2006. The year was selected based on the availability of POLDER aerosol products from the SRON retrieval algorithm."

TableS 1. List of selected meteorological and aerosol options of ECHAM-HAM used for the experiments.

| Description (Reference) | Model Option |
|---|---|
| Horizontal resolution of 1.875°, corresponding to 192 x 96 grid cells. For RESLOW only 3.75° | hres = T63 |
| Vertical resolution of 31 hybrid sigma pressure levels up to 10hPa | vres = L31 |
| Cumulus cloud convection scheme (Nordeng et al., 1994) | iconv = 1 |
| Sub-grid-scale stratiform clouds scheme (Sundqvist et al., 1989) | icover = 1 |
| Rapid Radiation Transfer Model for General circulation models (RRTM-G; Iacono et al., 2008) | - |
| Land surface model JSBACH (Reick et al., 2013) | - |
| Boundary layer parameterization (Stevens et al., 2013 and reference therenin) | - |
| Nudge vorticity, divergence, temperature and surface pressure to ERA5 reanalysis | - |
| Dust emission scheme (Stier et al., 2005) with updated East Asia soil properties | ndust = 4 |

| | |
|---|---|
| Sea salt emission scheme (Long et al., 2011) | nseasalt = 7 |
| Air-sea exchange parameterization for DMS emissions (Nightingale, 2000) | npist = 3 |
| Kappa-Koehler theory for aerosol water growth (Petters and Kreidenweis, 2007) | nwater = 1 |
| Size depended in-cloud and below-cloud scavenging (Tegen et al., 2019 and reference therein) | nwetdep = 3 |
| Enable interactive dry deposition scheme (Tegen et al., 2019 and reference therein) | ndrydep = 1 |
| Enable radiatively active aerosol | naerorad = 1 |

Maybe they authors could think of revising the title inserting also the term of assimilation.

The title has been changed to: "Assimilation of POLDER observations to estimate aerosol emissions"

Caption of Figure S1: I think it is wrongly written (f) sulphur dioxide (SO2). It should rather be WAT. Furthermore, the acronym WAT should be also defined in the caption and could be introduced in the text of the manuscript when the water uptake of soluble species (resulting to high AOD values) is discussed (e.g. lines 575-584).

The caption and the figure has been updated and corrected: "FigureS 1. Optical depth at 550nm of $CTL_{ECHAM}$ for (a) dust (DU), (b) sea salt (SS), (c) organic carbon (OC), (d) black carbon (BC), (e) sulphates ($SO_4$) and (f) water condensed on the surface of aerosol particles (WAT). The global contribution of each species to the total aerosol optical depth at 550nm is depicted at the right bottom corner. Third and fourth row depicts the contribution of each species to total aerosol optical depth at 550nm in each pixel."

I addition, we explain the abbreviation "WAT" in the first reference of FigureS 1 on section 4.1: "The $CTL_{ECHAM}$ $AOD_{550}$ per species along with the optical depth due to condensed water on the surface of aerosol particles (WAT) is depicted in FigureS 1."

Caption of Figure S8: For consistency with the text, it should be noted as "CTLERA5" than simply "ERA5".

In FigureS 8a the ERA relative humidity is depicted, which is used to compute aerosol water growth in the experiments $CTL_{ERA5}$. The caption has been corrected to make that clear: "FigureS 8. The relative humidity of (a) ERA5 used for aerosol water growth in $CTL_{ERA5}$, (b) $CTL_{ECHAM}$ and the difference (c) $CTL_{ECHAM}$ − ERA5 for 2006 at 800hPa. The global mean, the global mean error (ME) and the global mean absolute error (MAE) is depicted at the right bottom corner of each plot."

line 20: I would suggest " over isolated island sites at the ocean" instead of "over isolated island sites over the ocean".

Corrected as suggested.

line 24: Define at some place the acronyms such as GFAS.

The full name along with the abbreviation has been added to the first reference of GFAS in the abstract: "The biomass burning changes (based on POLDER) can be used as alternative biomass burning scaling factors for the Global Fire Assimilation System (GFAS) inventory distinctively estimated for organic carbon (2.93) and black carbon (1.90), instead of the recommended scaling of 3.4 (Kaiser et al. 2012)."

line 48 and line 50: You may delete "note that" in both sentences.

Corrected as suggested.

line 50: It should read "(from 1m to several km)" instead of (about 1m to several km).

Corrected as suggested.

line 52: "Emissions from biomass burning" instead of "Emissions from biomass burning emissions"

Corrected as suggested.

Lines 54-56: Please define at some place the acronyms such as GFED4, FINN1.5, QFED2.4, FEER1.0 and GFAS.

The full name along with the abbreviation was added in the introduction where this datasets are mentioned: "Emissions from biomass burning are based on satellite measurements that are related to burned area and use emission factors to convert the burned dry matter into emissions of aerosol and gas species (Global Fire Emissions Database v4 (GFED4); Van Der Werf et al., 2017), active fire count (Fire INventory from NCAR v1.5 (FINN1.5); Wiedinmyer et al., 2011) or fire radiative power (Quick Fire Emissions Dataset v2.4 (QFED2.4); Darmenov & da Silva, 2015, Fire Energetics and Emissions Research version v1.0 (FEER1.0); Ichoku & Ellison, 2014 and Global Fire Assimilation System (GFAS); Kaiser et al., 2012)."

Line 62: Is the term "diversity" used throughout the manuscript, the proper or the common word to express the ratio of the standard deviation to the mean. In many studies the word "range" is commonly used.

Thank you for your comment. The range can be misinterpreted as the difference between the maximum and the minimum value in a distribution. In this study we defined relative diversity of a distribution as the ratio of standard deviation to the mean, as in the study Schutgens et al., (2020).

Schutgens, N., Sayer, A. M., Heckel, A., Hsu, C., Jethva, H., de Leeuw, G., Leonard, P. J. T., Levy, R. C., Lipponen, A., Lyapustin, A., North, P., Popp, T., Poulsen, C., Sawyer, V., Sogacheva, L., Thomas, G., Torres, O., Wang, Y., Kinne, S., Schulz, M., and Stier, P.: An AeroCom–AeroSat study: intercomparison of satellite AOD datasets for aerosol model evaluation, Atmos. Chem. Phys., 20, 12431–12457, https://doi.org/10.5194/acp-20-12431-2020, 2020.

line 70: I would suggest "at least" instead of "at best".

Corrected as suggested.

line 72: I would suggest the plural "precursors" instead of "precursor".

Corrected as suggested where applicable throughout the document.

line 74: Maybe "assessed" or "found" instead of "used".

Corrected "used" to "found"

line 79: I suggest " ...similar information and methods which they are not ..." instead of "... similar information and methods and are not ..".

Corrected as suggested.

lines 119-121: The sentence is not clear and it needs some rephrasing.

Thank you for noting that, the sentence was indeed unclear. It was rephrased as:

"Although this study was very insightful, the discretization of scattering enhancement factor based on RH could correspond to a diverse aerosol load for each model. The low and high RH conditions may have occurred in different times and dates for every model, as well as for the observations."

line 273: It is not clear what is the setup of the $DAS_{ERA5}$ experiment? Is it the data assimilation experiment in ECHAM-HAM using relative humidity for ERA5? Please clarify in the text.

Thank you for bringing this up. A short description was added in the subsection "3.2 Experimental Setup" as depicted below. In addition, the title of subsection was corrected from "3.2 The Local Ensemble Transform Kalman Smoother" to "3.2 Experimental Setup"

"$CTL_{ERA5}$ quantifies the effect of the underestimated relative humidity in ECHAM compared to ERA5 on aerosol optical properties. $CTL_{ERA5}$ uses the relative humidity of ERA5 for aerosol water uptake. Note that this modification affects only the simulated aerosol optical properties in ECHAM-HAM, while the simulated water cycle (precipitation and evaporation) of the model remains unaltered. A data assimilation experiments based on this new $CTL_{ERA5}$ setup was conducted named $DAS_{ERA5}$ in order to quantify the effect of overestimated relative humidity profile to the aerosol emission estimation."

line 298: "along with the MAE" instead of "along the MAE".

Corrected as suggested.

line 298: " 3-hourly differences between the Experiments – POLDER" . It is better to be more specific. e.g. 3-h differences of $CTL_{ECHAM}$-POLDER and $DAS_{ECHAM}$-POLDER.

Corrected as suggested.

lines 531-533: Considering the complexity of the loss and production processes that control the SO2 and $SO_4$ fate in the atmosphere mentioned in this sentence, it could be nice to have a link to section 4.4.2 that you discuss these processes (e.g. as discussed in Section 4.4.2).

Thank you for your suggestion, a sentence was added that refers to the subsection 4.4.2 at that point.

"Thus, highlighting that inter-model differences in $SO_4$, may be caused primarily by differences in gain and loss processes rather than differences in the primary $SO_2$ emissions. The production and loss processes of $SO_4$ are discussed in more detail in subsection 4.4.2."

line 544: It should be "Figure 10" instead of "Figure 9".

Corrected.

line 579: " ...matches the underestimation of RH by ECHAM-HAM (Figure 10c) while ..." I am rather confused here with the underestimation. Do you mean the small underestimation over ocean (Figure 10 c) below 500 m? Above this level there is clear overestimation of RH.

Thank you for noting this. I was referring to the clear overestimation, the sentence has been corrected.

"Consequently, over ocean aerosol extinction profile differences (Figure 11c) matches the overestimation of RH by ECHAM-HAM (Figure 10c) while over land this is not the case (Figure 11b and Figure 10b)."

---

## Author Comment (AC2)

**Response to Referee #2 for the manuscript: "Assimilation of POLDER observations to estimate aerosol emissions"**

Dear Referee #2,

Thank you for reviewing our manuscript. Your comments help us to improve and define better some aspects of our work. Below you can find our point-by-point responses to all of your comments.

Best regards,
On behalf of all co-authors
Athanasios Tsikerdekis

**Format**
Questions
Responses
"Quotes from the manuscript and revised or added text."

**Comments**

I appreciate the authors discuss explicitly the values of annual emissions of global aerosol species and intercompare with other studies. However, what is missing but important is the daily or monthly variations of emissions which is one of the superiorities of top-down technique. I would suggest some discussions in the present study or more explicit study in the future to investigate the capability to capture the daily or monthly emission variations and to intercompare the variations from top-down emission datasets, for example, some datasets used in Elguindi et al. (2020, 1029/2020EF001520).

Thank you for your comment. The focus of the present study was to compare the estimated aerosol emissions with other modeling and data assimilation studies (e.g. Figure 9). Most of these studies reported the mean annual global emissions, hence the results were presented in a similar way.

I strongly agree that the clear advantage of top-down techniques is to capture rapid changes in emissions in a daily temporal resolution. In addition, satellite observations can capture regional changes in emission activity and can correct the bottom-up emission inventories with this up-to-date information. We added the following paragraph at the end of the conclusions to promote a more in-depth analysis on a daily basis for future studies:

"The focus of the present study was to estimate new aerosol emissions based on POLDER, evaluate the results with independent observations and inter-compare the estimated emissions with prior modelling and data assimilation studies on a yearly basis (Tg yr$^{-1}$). Future studies should focus also on highlighting the daily and monthly variation that top-down techniques can offer, as well as to take advantage of the up to date information provided by satellite observations, to correct bottom-up emission inventories over regions where emission activity has changed (Elguindi et al., 2020)."

Elguindi, N., Granier, C., Stavrakou, T., Darras, S., Bauwens, M., Cao, H., Chen, C., Denier van der Gon, H.A.C., Dubovik, O., Fu, T.M., Henze, D.K., Jiang, Z., Keita, S., Kuenen, J.J.P., Kurokawa, J., Liousse, C., Miyazaki, K., Müller, J.-.-F., Qu, Z., Solmon, F. and Zheng, B. (2020), Intercomparison of Magnitudes and Trends in

Anthropogenic Surface Emissions From Bottom-Up Inventories, Top-Down Estimates, and Emission Scenarios. Earth's Future, 8: e2020EF001520. https://doi.org/10.1029/2020EF001520

Although English is not my first language, I find some typos and mistakes in the manuscript. I would suggest the authors need to correct and revise carefully in terms of writing.

Thank you for your suggestion. We carefully re-read the manuscript, corrected typos and improved readability. A detailed document with track changes is included with our responses.

Abstract L21: Why do not include AEROCOM II?

Thank you for you comment. Indeed, AEROCOM II could have been added, though the main publication of that dataset was focused mostly on the aerosol direct radiative effect and the aerosol radiative forcing (Myhre et al. 2013). Thus, we decided to focus on AEROCOM I (Textor et al. 2006) and AEROCOM III (Gliß et al., 2021) where the main publications of these experiments, discussed aerosol emission in more detail.

Gliß, J., Mortier, A., Schulz, M., Andrews, E., Balkanski, Y., Bauer, S. E., Benedictow, A. M. K., Bian, H., Checa-Garcia, R., Chin, M., Ginoux, P., Griesfeller, J. J., Heckel, A., Kipling, Z., Kirkevåg, A., Kokkola, H., Laj, P., Le Sager, P., Lund, M. T., Lund Myhre, C., Matsui, H., Myhre, G., Neubauer, D., van Noije, T., North, P., Olivié, D. J. L., Rémy, S., Sogacheva, L., Takemura, T., Tsigaridis, K., and Tsyro, S. G.: AeroCom phase III multi-model evaluation of the aerosol life cycle and optical properties using ground- and space-based remote sensing as well as surface in situ observations, Atmos. Chem. Phys., 21, 87–128, https://doi.org/10.5194/acp-21-87-2021, 2021.

Myhre, G., Samset, B. H., Schulz, M., Balkanski, Y., Bauer, S., Berntsen, T. K., Bian, H., Bellouin, N., Chin, M., Diehl, T., Easter, R. C., Feichter, J., Ghan, S. J., Hauglustaine, D., Iversen, T., Kinne, S., Kirkevåg, A., Lamarque, J.-F., Lin, G., Liu, X., Lund, M. T., Luo, G., Ma, X., van Noije, T., Penner, J. E., Rasch, P. J., Ruiz, A., Seland, Ø., Skeie, R. B., Stier, P., Takemura, T., Tsigaridis, K., Wang, P., Wang, Z., Xu, L., Yu, H., Yu, F., Yoon, J.-H., Zhang, K., Zhang, H., and Zhou, C.: Radiative forcing of the direct aerosol effect from AeroCom Phase II simulations, Atmos. Chem. Phys., 13, 1853–1877, https://doi.org/10.5194/acp-13-1853-2013, 2013.

Textor, C., Schulz, M., Guibert, S., Kinne, S., Balkanski, Y., Bauer, S., Berntsen, T., Berglen, T., Boucher, O., Chin, M., Dentener, F., Diehl, T., Easter, R., Feichter, H., Fillmore, D., Ghan, S., Ginoux, P., Gong, S., Grini, A., Hendricks, J., Horowitz, L., Huang, P., Isaksen, I., Iversen, I., Kloster, S., Koch, D., Kirkevåg, A., Kristjansson, J. E., Krol, M., Lauer, A., Lamarque, J. F., Liu, X., Montanaro, V., Myhre, G., Penner, J., Pitari, G., Reddy, S., Seland, Ø., Stier, P., Takemura, T., and Tie, X.: Analysis and quantification of the diversities of aerosol life cycles within AeroCom, Atmos. Chem. Phys., 6, 1777–1813, https://doi.org/10.5194/acp-6-1777-2006, 2006.

L52: Emissions from biomass burning or Biomass burning emissions

Corrected as suggested: "Emissions from biomass burning are estimated based on satellite…"

L52: … are estimated based on satellite measurements …

Corrected as suggested: "Emissions from biomass burning are estimated based on satellite…"

L72: -> differences ... are ...

Corrected as suggested: "The anthropogenic emissions differences between inventories for aerosol or aerosol precursor are considerably lower than the one of natural emissions."

L97: Zhang et al. (2015, 10.5194/acp-15-10281-2015) assimilate OMI AAOD to estimate BC emission over East Asia.

Thank you for proposing this study, it is an innovative work and one of the first to assimilate aerosol absorption observations. A reference was added: "Note that most of these studies estimate new emissions based on the assimilation of Aerosol Optical Depth (AOD), some may include also Ångström Exponent (AE), while very few assimilate absorption observations, like Absorption Aerosol Optical Depth (AAOD) or Single Scattering Albedo (SSA) (Zhang et al., 2015; Chen et al., 2018, 2019)."

Zhang, L., Henze, D. K., Grell, G. A., Carmichael, G. R., Bousserez, N., Zhang, Q., Torres, O., Ahn, C., Lu, Z., Cao, J., and Mao, Y.: Constraining black carbon aerosol over Asia using OMI aerosol absorption optical depth and the adjoint of GEOS-Chem, Atmos. Chem. Phys., 15, 10281–10308, https://doi.org/10.5194/acp-15-10281-2015, 2015.

L155: The POLDER observations... Actually, you are using POLDER aerosol products derived from POLDER observations.

Corrected as suggested: "The aerosol products derived from POLDER observations that were"

L170: Why do you use AE550-865, instead of AE440-865 which should be more proper over desert region?

Thank you for your suggestion. We chose to assimilate $AE_{550-865}$ to be consistent with our preceding works (Tsikerdekis et al., 2021 and 2022). Note also that the $AE_{550-865}$ of POLDER over the desert region (e.g. Sahara) was found to be bias high (>0.5) when compared to AERONET $AE_{550-865}$, hence the $AE_{550-865}$ cases over Sahara has been removed. In the future we would like to conduct more sensitivity studies on aerosol emissions with different AE wavelength ranges, like the sensitivity studies conducted in Tsikerdekis et al. (2021).

Tsikerdekis, A., Schutgens, N. A., Fu, G., & Hasekamp, O. P. (2022). Estimating aerosol emission from SPEXone on the NASA PACE mission using an ensemble Kalman smoother: observing system simulation experiments (OSSEs). Geoscientific Model Development, 15(8), 3253–3279.

Tsikerdekis, A., Schutgens, N. A., & Hasekamp, O. P. (2021). Assimilating aerosol optical properties related to size and absorption from POLDER/PARASOL with an ensemble data assimilation system. Atmospheric Chemistry and Physics, 21(4), 2637–2674.

L199: ... based on Petters and Kreidenweis (2007).

Corrected as suggested.

L213: aerosol emission or aerosol emission fluxes? Please keep one consistently throughout the paper.

Thank you for this suggestion. We kept "aerosol emission" throughout the text.

L231: have you ever tried to understand the effects on derived emissions once you change the assimilation time window?

The $\Delta T = 2$days was selected due to time and computer resources constrains, hence its effect on the derived emissions was not studied with additional sensitivity experiments. The emissions are estimated in 2-day time windows, hence the assumption is that emissions do not change drastically over the course of 2-day time windows, as discussed on Section "3.1 The Local Ensemble Transform Kalman Smoother". The same was assumed in Schutgens et al. (2012). This may not be the case of course for some emission types such as dust, that can vary from day to day.

It is worth investigating the impact of that choice in a future study, thus the following has been added to the Conclusions: "In addition, our estimated emissions were based on a two-day timestep ($\Delta T=2$ days), hence a follow-up study could explore the impact of a lower timestep (e.g. $\Delta T=1$ day) to the estimated emissions."

Schutgens, N., Nakata, M., & Nakajima, T. (2012). Estimating aerosol emissions by assimilating remote sensing observations into a global transport model. Remote Sensing, 4(11), 3528–3543.

L257: R -> bolded R

Corrected as suggested.

Equations 1,2: Please use consistent font, italic, bold as in the main text for matrix and vector.

Corrected as suggested.

The subtitles of 3.1, 3.2 and 4.1 are the same, please correct them.

Thank you for noting that. The section 3.1 remained as is while the sections 3.2, 4.1 and 4.3 were corrected to:

"3.1 The Local Ensemble Transform Kalman Smoother"
"3.2 Experimental Setup"
"4.1 Evaluating model fields with POLDER, AERONET and MODIS observations"
"4.3 Global aerosol emissions and comparison with other studies"

The subtitles of 4.2 and 4.3 are the same, please correct.

Kept the subsection title as is for 4.2 and changed the title of 4.3 to:

"4.3 Global aerosol emissions and comparison with other studies"

Figure S1 (f): what do you mean of WAT AOD550 which accounts for 62% of total global AOD?

Thank you for bringing this up. WAT is an abbreviation for optical depth produced due to condensed water on the surface of aerosol particles (hygroscopic growth or wet growth). The global WAT $AOD_{550}$ contributes up to 62% of global $AOD_{550}$.

An explanation was added in subsection 4.1 for the WAT abbreviation where Figure S1 is mentioned for the first time: "The $CTL_{ECHAM}$ $AOD_{550}$ per species along with the optical depth due to condensed water on the surface of aerosol particles (WAT) is depicted in FigureS 1"

In addition, the caption of FigureS 1 has been corrected: "FigureS 1. Optical depth at 550nm of $CTL_{ECHAM}$ for (a) dust (DU), (b) sea salt (SS), (c) organic carbon (OC), (d) black carbon (BC), (e) sulphates (SO4) and (f) water condensed on the surface of aerosol particles (WAT). The contribution of each species to the total aerosol optical depth at 550nm is depicted at the right bottom corner."

Figure A1. POLDER uncertainty for AOD... It's better to indicate POLDER SRON product uncertainty.

Corrected as suggested.

Figure A1. In my opinion, the relative uncertainty for AE is a bit meaningless, because it may be dominant by the coarse mode cases where the AE is small and the relative uncertainty is high.

We agree, thus we did not calculate the uncertainty of $AE_{550-865}$ in relative terms but in $AE_{550-865}$ absolute terms as depicted in Figure A1 y axis. We did the same for SSA550 as well since

it is a ratio of AOD550 and AAOD550. The uncertainty was calculated in relative terms only for AOD550 (divided by AERONET AOD550 in each case). We slightly modified the caption to highlight this better:

"FigureA 1. POLDER SRON product uncertainty for AOD550, AE550-865 and SSA550 based on an AERONET evaluation for several POLDER AOD550 bins. Red and blue lines depict the uncertainty of over land and over ocean retrievals respectively (left axis). The respective colored bars illustrate the number of collocated POLDER and AERONET retrievals were used to calculate the observables uncertainty in each AOD550 bin (right axis) and N depicts the total number. Note that only AOD550 uncertainty was estimated in relative terms, by dividing with AERONET AOD550. The AOD550 and AE550-865 evaluation is based on AERONET Version 3 Direct Sun Algorithm Level 2.0, while the AAOD550 and SSA550 evaluation is based on AERONET Version 3 Direct Sun and Inversion Algorithm Level 1.5."

L285: how do you deal with dust bins and emission fraction of each dust bins?

Thank you for your question. The estimated emissions (state vector) in our system are distinctive for each aerosol species, each size mode (not bins for ECHAM-HAM) and in certain cases distinctive for some emission types (e.g. biomass burning sources), as shown in Table 1. Specifically for dust, we estimate distinctively the emissions for the Accumulation and the Coarse dust insoluble modes emitted in the model. This is explained in the subsection "3.1 The Local Ensemble Transform Kalman Smoother":

"where xb is the background state vector and includes emission perturbations for each species (DU, SS, OC, BC and SO4). Different perturbations are used for each optically relevant mode (Aitken, Accumulation, Coarse) and biomass burning (BB) or fossil fuel (FF) contributions. Specifically, the emissions that are distinctively perturbed and estimated (11 in total) by the assimilation system are shown in Table 1."

Figure S2: could you explain how can you obtain POLDER uncertainty of AE? why is it 'no value' over Sahara region?

The calculation of AE uncertainty was calculated by collocating POLDER and AERONET observations, partitioning these pairs based on POLDER AOD550 groups (as shown in Figure A1) and calculate the standard deviation of the differences POLDER AE – AERONET AE (y axis of FigureS 1) for each POLDER AOD group (x axis of FigureS. 1). This was done separately for retrievals over the ocean and land. An explanation was added in the Appendix A1:

"The uncertainty of POLDER observations is estimated by evaluating it with AERONET for predefined POLDER AOD bins. Uncertainty is defined as the standard deviation of the differences between POLDER and AERONET observations, for different POLDER AOD bins. For AOD only, a relative uncertainty was used. by dividing with AERONET AOD in each case. FigureA 1 depicts the uncertainty for AOD, AE and SSA."

In addition, the Sahara region AE was not assimilated and was not used for the evaluation because of a high overestimation of POLDER AE compared to AERONET AE as noted in subsection 2.1 Aerosol Observations (POLDER) and Appendix A:

"Note that POLDER AE550-865 over Sahara is biased high based on AERONET, thus these observations were not assimilated (see Appendix A)."

"Further, we found that over Sahara $AE_{550-865}$ is overestimated by POLDER by 0.524, thus these observations were not used in the assimilation."

L308-309: what cause high AAOD over Australia coastline?

The high AAOD550 over the northern and western coast of Australia are probably caused by retrieval errors. The following was added to the text:

"Further, high $AAOD_{550}$ values are depicted over the northern and western coastline of Australia, which probably is a product of retrieval errors."

Figure S3: the global mean value does not appear at the left bottom corner.

Thank you for the note. The caption of FigureS 3 was corrected, and figure was updated: "FigureS 3. Absorption aerosol optical depth at 550nm (AAOD550) of CTLECHAM for (a) black carbon, (b) dust and (c) organic carbon (first row), along with the contribution of each species to the total absorption aerosol optical depth at 550nm in each pixel (second row). The percentage in the bottom left corner indicates the global contribution of each species to AAOD550."

L320-321: in this paragraph, you talk about AAOD, why you report MAE for SSA at the end, instead of MAE for AAOD?

Thank you for noting this, it has been corrected to: "The global MAE for $AAOD_{550}$ is reduced from 0.0106 in CTL_ECHAM to 0.0077 in DAS_ECHAM."

L355-357: it's interesting to know the effect on absorption is negative if only AOD is assimilated. Do you have any explanations?

This has been demonstrated in our previous work (Tsikerdekis et al., 2021) where the state vector was aerosol mixing ratio per aerosol species. When only AOD was assimilated, the aerosol mixing ratio changed based only on the extinction of each species, with no information on the absorption. In that case, absorbing aerosol species (e.g. see Figure S3 BC, DU and OC of the current paper as a reference) were adjusted but ended up overestimated AAOD more since their mixing ratio was not constrained based on their AAOD (see Figure 9h on Tsikerdekis et al., 2021). When both AOD and AAOD were assimilated, the aerosol

mixing ratio adjusted per species based on both their extinction and absorption, providing better results for both AOD and AAOD (see Figure 5j and Figure 9j on Tsikerdekis et al., 2021). The negative effect of assimilating only AOD can be more prominent in areas where a mixture of scattering and absorbing aerosol is found (e.g. biomass burning plumes that contain OC and BC aerosol). In conclusion, theoretically the AOD-only assimilation may lead to an overestimation or underestimation of AAOD, depending on if AOD was overestimated or underestimated in the control experiment.

L361: the latest C6 and C6.1 products should be better than C5 for comparison. In addition, why do you use DT only, you may find more insights from DT+DB combined dataset over bright surface?

Indeed, the latest versions of MODIS (C6 and C6.1) have improved in a lot of aspects compare to the C5 collection. We evaluated against C5 since we had a specialized version of that product which was corrected based on four years of AERONET observations as noted in the text: "Specifically, we use a specialized version of MODIS designed for assimilation, which was corrected based on four years of AERONET observations (Hyer et al., 2011; Shi et al., 2011; J. Zhang & Reid, 2006)."

In addition, this version of MODIS was used for evaluation purposes in our preceding work (Tsikerdekis et al., 2021).

L390: do you have high confidence of POLDER AE over ocean where the aerosol load is generally low?

As shown in Figure A1, the lower POLDER AOD gets the higher AE uncertainty is. For very low POLDER AOD (between zero and 0.1) the uncertainty of AE 0.5. Based on that there is low confidence of AE over ocean for low aerosol load cases. On the other hand, for cases above 0.3 AOD, the uncertainty of AE is quite low (0.2).

L407: AEROCOM phase II?

Responded above for the question Abstract L21: Why do not include AEROCOM II?

L409: The latest study by Chen et al. (2022, 10.1038/s41467-022-35147-y) report global values from 2006-2011, and the emission for reference year 2010 is reported in Chen's 2019 paper (10.5194/acp-19-14585-2019).

Thank you for that correction. The reference has been corrected as: "Note that the annual mean emissions for some studies may be regional and not global estimates (e.g. Chen et al., 2019; Escribano et al., 2017) and also may not refer to year 2006, which is the reference year for our study."

A reference was added to the latest Chen et al. (2022) study in the introduction. In addition, Figure 9 was updated in order to include Chen et al. (2022) emission estimates for dust, organic aerosol and black carbon.

[Figure]

Figure 9. Global aerosol emissions of 2006 for Dust (DU), Sea Salt (SS), Organic Aerosol (OA), Black Carbon (BC), Sulfur Dioxide (SO2) and total deposition of Sulfates (SO4) (Tg yr-1). The percentage change of the estimated emissions over DASECHAM is estimated based on the emissions of CTLECHAM respectively.

Circles depict the reported emissions from other studies. Diamond depicts the sensitivity study in Textor et al. (2007) which is explained in the text. Triangles illustrate the emissions estimated from past data assimilation studies. OA is estimated by multiplying the emissions of OC with 1.4 for the experiments of this study, as well for the reported emissions in Schutgens et al. (2012), Chen et al. (2019) and Chen et al. (2022). SO4 total deposition is used as a proxy for SO4 pseudo-emissions. SO2 emissions for Chen et al. (2019) and Huneeus et al. (2012) were reported in Tg S yr-1, thus they were multiplied with 2 in order to be converted in Tg SO2 yr-1. The asterisk symbol on some studies indicate that the emissions reported are regional and not global. The yearly emissions from Schutgens et al. (2012) are an extrapolation of a single month's (January) experiment. The two Kok et al. (2021) estimates refer to emissions for DU particles up to 10μm (low estimate) and up to 20μm (high estimate) in geometric diameter (see text for more details).

Chen, C., Dubovik, O., Schuster, G.L. et al. Multi-angular polarimetric remote sensing to pinpoint global aerosol absorption and direct radiative forcing. Nat Commun 13, 7459 (2022).

L419: permitted -> emitted?

Corrected according to suggestion: "The emissions of these species are highly dependent on the simulated aerosol size range of each model, wind distribution in each model as well as the activation areas, where dust can be emitted, hence the emissions differ a lot from model to model (Wu, 2020)."

L426: dust emission is highly relying on the particle size range that you are modeling, could you explicit your setup of dust bins?

The vertical emissions fluxes are integrated over 192 dust size classes (ranging from 0.2 to 1300μm) and are summed up into the unimodal lognormal particle size distributions of ECHAM-HAM (Cheng et al. 2008), which are called modes.

Specifically for dust, they are summed into two modes: the accumulation insoluble (AI) and coarse insoluble (CI) modes. Each mode is characterized by the number concentration and the mass concentration by species. Aerosol number and mass are used in order to calculate the median radius for each mode. The mode width (standard deviation of the lognormal distribution) is assumed and fixed as equal to 1.59 for the accumulation mode and 2.00 for the coarse mode (Tegen et al., 2019). The mass and number emission of these two modes (AI and CI) are used as state vector in our data assimilation system to distinctively estimate new emissions for each of these modes.

The median radius of the CI mode (that is consisted only by dust insoluble particles) is about 0.37μm over the Sahara. Considering that the uni-modal log-normal distribution of the coarse mode has a standard deviation of 2, we can determine the size of particles up to a certain percentile of the distribution. For the 90% percentile a particle radii is 4.8μm, for 95% is 9.95μm and for 99% is 38.95μm.

As correctly pointed out, the dust (as well as sea salt) total emissions rely on the modelled particle size range. Potentially, adding a super-coarse mode in ECHAM-HAM for dust (and sea salt) could increase the estimate emissions by our system. This hypothesis is supported by the recent study by Kok et al. (2021) that shows very large differences when considering emissions up to 10μm compared to 20μm (see Figure 9 and discussion 4.3.1 Dust emissions).

A sentence was added to the subsection 4.3.1 Dust emissions: "The contribution of emitted particles between 10μm and 20μm to the total dust emissions was close to 65%, but the contribution to the total AOD550 in the same size range was about 7%. Based on this, results, the inclusion of a super-coarse insoluble mode in ECHAM-HAM will increase total emissions and AOD550 over dust areas as well as the estimated emissions by our data assimilation system."

As well as in the conclusions: "The new dust emissions are very close to the ensemble median of AEROCOM, and match quite well the estimated emissions reported by other data assimilation studies (Hueneeus et al., 2012). However, the addition of a super-coarse mode for dust could increase the modelled dust emissions as well as the estimated dust emissions from our data assimilation system (Kok et al., 2021)."

Cheng, T., Peng, Y., Feichter, J., and Tegen, I.: An improvement on the dust emission scheme in the global aerosol-climate model ECHAM5-HAM, Atmos. Chem. Phys., 8, 1105–1117, https://doi.org/10.5194/acp-8-1105-2008, 2008.

Tegen, I., Neubauer, D., Ferrachat, S., Siegenthaler-Le Drian, C., Bey, I., Schutgens, N., Stier, P., Watson-Parris, D., Stanelle, T., Schmidt, H., Rast, S., Kokkola, H., Schultz, M., Schroeder, S., Daskalakis, N., Barthel, S., Heinold, B., and Lohmann, U.: The global aerosol–climate model ECHAM6.3–HAM2.3 – Part 1: Aerosol evaluation, Geosci. Model Dev., 12, 1643–1677, https://doi.org/10.5194/gmd-12-1643-2019, 2019.

Kok, J. F., Adebiyi, A. A., Albani, S., Balkanski, Y., Checa-Garcia, R., Chin, M., Colarco, P. R., Hamilton, D. S., Huang, Y., Ito, A., & others. (2021). Improved representation of the global dust cycle using observational constraints on dust properties and abundance. Atmospheric Chemistry and Physics, 21(10), 8127–8167.

L441: some of the factors … are …

Corrected as suggested.

L611: -> global SS emissions

Corrected as suggested.

L710: is there any comment on the treatment of absorbing organic carbon?

Thank you very much for giving us a chance to elaborate further on this topic. Absorbing organic carbon, known also as brown carbon, is not simulated by ECHAM-HAM. As discussed in Chen et al. (2019), the absence of brown carbon may lead to an overestimation of BC emissions over biomass burning regions, when AAOD is underestimated in the control experiment. The contribution of organic carbon to Absorbing Aerosol Optical Depth (AAOD) is very low in the model (globally 4% of total AAOD, see updated FigureS 3 below). OC AAOD contribution total AAOD may reach 10% to biomass burning regions in the Tropics (South America, Africa, Indonesia), hence the emissions estimation for OC is mainly controlled by AOD. Recent studies suggested that the contribution of brown carbon to AAOD may be up to 40% (Zhang et al., 2021).

The following sentence was added to the conclusions: "It is noted that the absorbing organic aerosol (known also as brown carbon), which strongly absorb radiation in the ultraviolet wavelengths, are not considered. The OC AAOD contribution to total AAOD in our experiments is about 10% over the biomass regions in the Tropics (South America, Africa and Indonesia), while the rest 90% is contributed by BC AAOD. The exclusion of brown carbon, may lead to an overestimation of the BC emissions by the data assimilation system, as discussed also in Chen et al. (2019). Brown carbon is a topic of ongoing research and recent studies suggested that may contribute up to 40% to the total AAOD (Zhang et al. 2021)."

Chen, C., Dubovik, O., Henze, D. K., Chin, M., Lapyonok, T., Schuster, G. L., Ducos, F., Fuertes, D., Litvinov, P., Li, L., & others. (2019). Constraining global aerosol emissions using POLDER/PARASOL satellite remote sensing observations. Atmospheric Chemistry and Physics, 19(23), 14585–14606.

Y. Zhang, Y. Peng, W. Song, Y.L. Zhang, P. Ponsawansong, T. Prapamontol, Y. Wang (2021). Contribution of brown carbon to the light absorption and radiative effect of carbonaceous aerosols from biomass burning emissions in Chiang Mai, Thailand. Atmos. Environ., 260, Article 118544, 10.1016/j.atmosenv.2021.118544

[Figure]

**FigureS 1. Absorption aerosol optical depth at 550nm (AAOD$_{550}$) of CTL$_{ECHAM}$ for (a) black carbon, (b) dust and (c) organic carbon (first row), along with the contribution of each species to the total absorption aerosol optical depth at 550nm (second row). The percentage in the bottom corner indicates the global contribution of each species to AAOD$_{550}$.**